# Relation-Aware Diffusion for Heterogeneous Graphs with Partially Observed Features

**Daeho Um**[*]
AI Center, Samsung Electronics
daeho.um@samsung.com

**Yoonji Lee**
Samsung Electronics
yj811.lee@samsung.com

**Jiwoong Park**
Department of Electrical and Computer Engineering
Texas A&M University
ptywoong@gmail.com

**Seulki Park**
University of Michigan
seulki@umich.edu

**Yuneil Yeo**
Department of Civil and Environmental Engineering
UC Berkeley
yuneily@berkeley.edu

**Seong Jin Ahn**
KAIST
sja1015@kaist.ac.kr

## Abstract

Diffusion-based imputation methods, which impute missing features through the iterative propagation of observed features, have shown impressive performance in homogeneous graphs. However, these methods are not directly applicable to heterogeneous graphs, which have multiple types of nodes and edges, due to two key issues: (1) the presence of nodes with undefined features hinders diffusion-based imputation; (2) treating various edge types equally during diffusion does not fully utilize information contained in heterogeneous graphs. To address these challenges, this paper presents a novel imputation scheme that enables diffusion-based imputation in heterogeneous graphs. Our key idea involves (1) assigning a *virtual feature* to an undefined node feature and (2) determining the importance of each edge type during diffusion according to a new criterion. Through experiments, we demonstrate that our virtual feature scheme effectively serves as a bridge between existing diffusion-based methods and heterogeneous graphs, maintaining the advantages of these methods. Furthermore, we confirm that adjusting the importance of each edge type leads to significant performance gains on heterogeneous graphs. Extensive experimental results demonstrate the superiority of our scheme in both semi-supervised node classification and link prediction tasks on heterogeneous graphs with missing rates ranging from low to exceedingly high. The source code is available at https://github.com/daehoum1/hetgfd.

## 1 Introduction

Missing data is a prevalent problem in the real world, caused by various factors including measurement failures and cost constraints during data collection (Allison, 2009). This poses a challenge for various machine learning techniques, as they typically assume complete (fully observed) input data. To address this problem, traditional imputation methods such as zero or mean imputation are widely utilized to fill in missing values (Armitage et al., 2015). However, in situations with high missing rates, traditional imputation methods show limitations in maintaining the performance of machine learning techniques in downstream tasks (Rossi et al., 2022). Moreover, missing data problem becomes more challenging when handling graph-structured data, which contain internal relations.

To address this problem in graph-structured data, recently, imputation methods based on graph diffusion (Rossi et al., 2022; Um et al., 2023; 2025b) have been proposed, which iteratively propagate observed features to missing features without training neural networks. These diffusion-based methods have attracted great attention due to their various advantages: exceptional performance in downstream tasks, fast imputation time, and being agnostic to downstream network design. Diffusion-

---

[*]Corresponding author

based methods assume that given data are homogeneous graphs, where all nodes are of the same type and all edges represent the same types of relations. However, there are various objects and relations in real-world scenarios. Heterogeneous graphs, containing multiple types of nodes and edges, have been widely studied due to their ability to model the diverse and complex nature of real-world scenarios, such as wireless networks with various devices and link types (Zhang et al., 2019).

Although both heterogeneous graphs and the problem of missing data hold important positions in real-world applications, imputation for heterogeneous graphs is very challenging due to the complex relations between various objects. In most real-world heterogeneous graph datasets, only nodes of a certain type possess features. Figure 1 demonstrates an example of a real-world heterogeneous graph. As shown in the figure, only 'Movie' nodes have features (*i.e.*, are attributed) while 'Actor' and 'Director' nodes do not have features (*i.e.*, are non-attributed). However, in real-world heterogeneous graphs, missing data can occur even within features of attributed nodes as 'Movie' nodes in Figure 1.

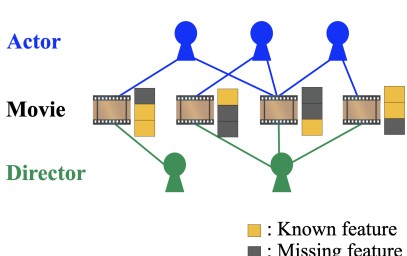

Figure 1: An example of a real-world heterogeneous graph.

While several studies have been proposed to address undefined features for non-attributed nodes (Jin et al., 2021; Wang et al., 2022), they are based on the strong assumption that all features in attributed nodes are fully observed. To the best of our knowledge, missing feature imputation within attributed nodes is underexplored. We tackle this problem in heterogeneous graphs, where both undefined features for non-attributed nodes and missing features within attributed nodes coexist, presenting a challenging but practical scenario.

When attempting to apply diffusion-based methods (Rossi et al., 2022; Um et al., 2023; 2025b) to missing feature imputation within attributed nodes on heterogeneous graphs, there are two challenging issues. (1) Attributed nodes often lack direct connections to each other, hindering imputation via diffusion. Since diffusion-based impute methods fill in missing values by aggregating features in neighboring nodes, missing features cannot be filled in without direct connections among attributed nodes. (2) In a heterogeneous graph, each edge type represents a distinct relation. For instance, in Figure 1, an 'Actor' and a 'Movie' node are connected via an 'is starred in' relation, while a 'Movie' node and a 'Director' node are connected via an 'is directed by' relation. Thus, simply treating all edge types equally cannot fully utilize the relational information contained in heterogeneous graphs.

To tackle these challenges in heterogeneous graphs, we present HetGFD (Heterogeneous Graph Feature Diffusion), a novel diffusion-based imputation method for heterogeneous graphs. Our key idea involves (1) generating virtual features for non-attributed nodes and (2) ranking edge types according to a new criterion called edge-type-wise homophily. For issue (1), virtual features on non-attributed nodes are initialized to zero values, and enable diffusion-based imputation by acting as real features. To address issue (2), during the diffusion process, we leverage our proposed edge-type-wise homophily to enable relation-aware distance encoding, leading to relation-aware diffusion. Through experiments, we verify that these methods successfully maintain their effectiveness in heterogeneous graphs. Furthermore, extensive experimental results demonstrate the superiority of our HetGFD over state-of-the-art methods in both semi-supervised node classification and link prediction tasks on benchmark datasets. We also show the applicability of HetGFD to the biomedical domain. We confirm that HetGFD performs well even with an extreme missing rate of 99.5%.

Our key contributions are summarized as follows: 1) To the best of our knowledge, this work is the first attempt to leverage diffusion-based feature imputation for heterogeneous graphs and to design relation-aware distance encoding. 2) Our virtual feature scheme enables the use of diffusion-based imputation methods and successfully transfers their effectiveness to the heterogeneous graph domain. 3) We define a new measure, *edge-type-wise homophily*, to adjust the influence of each edge type during diffusion. By utilizing edge-type-wise homophily, HetGFD, tailored to heterogeneous graphs, outperforms state-of-the-art methods across various domains, including the biomedical domain, by treating multiple edge types differently.

## 2 RELATED WORK

**Handling Missing Features in Graphs.** The problem of missing data has been extensively studied in the literature (Allison, 2009; Loh & Wainwright, 2011). However, in cases where the rate of missing features is substantial, accurate reconstruction becomes challenging. Besides, achieving

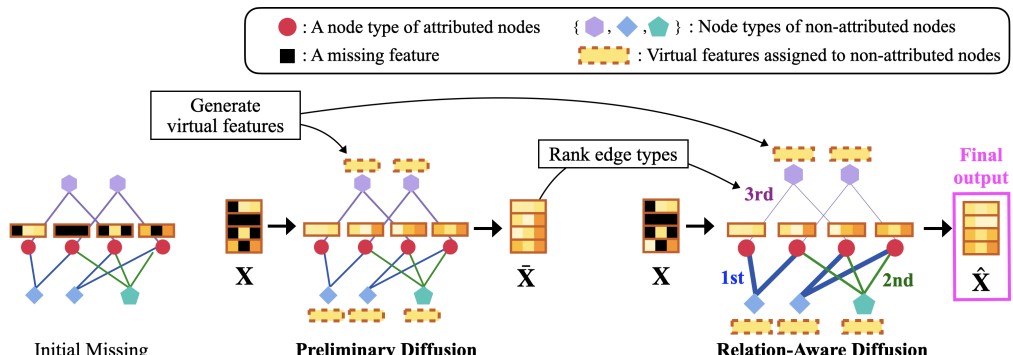

Figure 2: A brief overview of Heterogeneous Graph Feature Diffusion (HetGFD). Virtual features generated for non-attributed nodes enable diffusion-based imputation. Since the calculation of edge-type-wise homophily requires a complete feature matrix without missing features, we first impute missing features via the preliminary diffusion stage. Subsequently, using these pre-imputed features, edge types are ranked according to edge-type-wise homophily. A final output is obtained via relation-aware diffusion where the edge-type ranking plays an important role.

accurate reconstruction does not necessarily guarantee high performance in downstream tasks (Um et al., 2023). Thus, recent research directions of feature imputation in graphs focus on enhancing performance on downstream tasks rather than achieving perfect restoration. While GCN (Kipf & Welling, 2016a)-variant methods (Jiang & Zhang, 2020; Taguchi et al., 2021) utilize new GNN architectures to enhance the performance in downstream tasks with missing features, diffusion-based methods (Rossi et al., 2022; Um et al., 2023) (different from generative diffusion models (Ho et al., 2020; Dhariwal & Nichol, 2021; Yeo & Um, 2025)) have significantly improved the performance by imputing missing features through the diffusion of known features. However, as most methods for addressing missing features in graphs focus on homogeneous graphs, they cannot handle complex graphs with multiple node and edge types. Recently, several approaches have been proposed to learn representations for non-attributed nodes in heterogeneous graphs (He et al., 2022; Jin et al., 2021; Wang et al., 2022). However, these studies commonly assume that all nodes in an attributed node type have all their features intact, which can cover only limited missing scenarios. In contrast, we aim to handle the presence of missing features within attributed nodes, which makes the problem more challenging. While Zhang et al. (2023) is related to our work, it targets spatio-temporal graphs. Similarly, Gupta et al. (2023) is relevant; however, it does not consider multi-type nodes or edges.

**Heterogeneous GNNs.** While GNNs (Kipf & Welling, 2016a; Veličković et al., 2017; Lim et al., 2021; Um et al., 2025a) have proven to be a powerful approach for learning graph representations, most GNNs have primarily focused on homogeneous graphs. Hence there have been considerable efforts to extend the advantages of GNNs to heterogeneous graphs (Zhang et al., 2019; Wang et al., 2019; Fu et al., 2020; Hu et al., 2020b). For example, Heterogeneous Graph Transformer (HGT) (Hu et al., 2020b) adopts the architecture design of Transformer (Vaswani et al., 2017) to learn node representations before message passing. Additionally, Guo et al. (2023) proposes a meta-path-induced metric to quantify the degree of homophily. However, it can only deal with meta-paths between nodes of the same type. Unlike the homophily we define, this metric is measured based on pre-defined meta-paths rather than actual edges.

**Distance Encoding.** Based on feature homophily, we assign the importance of each feature during diffusion by measuring the shortest path distance between the feature and known features. Distance encoding for graphs generates auxiliary node features calculated by the distance from designated nodes to another node. These generated features are then utilized to perform a given task. For instance, nodes are assigned new features based on the distances to target nodes (Zhang & Chen, 2018; Zhang et al., 2021). Position-aware graph neural network (P-GNN) (You et al., 2019) leverages the distance between a given target node and sampled anchor node sets. To enhance the expressive power of graph neural networks, an additional node feature called distance encoding is proposed (Li et al., 2020). While distance encoding-based heterogeneous graph neural network (DHN) (Ji et al., 2021) extends distance encoding to heterogeneous graphs, DHN does not consider different edge types during the distance measurement. In contrast to previous approaches in distance encoding, we propose relation-aware distance encoding, considering multiple edge types.

## 3 PROBLEM SETTING

A heterogeneous graph is denoted by $\mathcal{G} = (\mathcal{V}, \mathcal{E}, \mathcal{T}, \mathcal{R})$ where $\mathcal{V} = \{v_i\}_{i=1}^N$ is the set of $N$ nodes, $\mathcal{E}$ is the set of edges, $\mathcal{T}$ is the set of node types, $\mathcal{R}$ is the set of edge types, and $|\mathcal{T}| + |\mathcal{R}| > 2$. To express nodes linked by an $r$-type edge, we let $(v_i, r, v_j)$ denote an $r$-type edge connecting $v_i$ and $v_j$, *i.e.*, $(v_i, r, v_j) \in \mathcal{E}$. We let $R = \{r_1, \ldots, r_{|\mathcal{R}|}\}$ to indicate each edge type in $\mathcal{R}$. $\mathcal{E}_r$ denotes the set of edges of type $r$ where $r \in \mathcal{R}$ so that $\cup_{r \in \mathcal{R}} \mathcal{E}_r = \mathcal{E}$. We introduce $\mathbf{A} \in \{0, 1\}^{N \times N}$ to denote an adjacency matrix. A representative dataset for a heterogeneous graph $\mathcal{G}$ is the DBLP dataset which consists of four node types (paper, author, term, and conference) and three edge types (paper-author, paper-term, and paper-conference).

The $t^+$-type nodes with partially known features are called *attributed nodes* and the other nodes where features are undefined are called *non-attributed nodes*. $\mathcal{V}^+$ denotes the set of attributed nodes. $\mathcal{V}^-$ denotes the set of non-attributed nodes. $N^+$ and $N^-$ denote $|\mathcal{V}^+|$ and $|\mathcal{V}^-|$, respectively, where $|\mathcal{V}|$ indicates the cardinality of a node set $V$. Letting $F$ be the number of feature channels of each node in $\mathcal{V}^+$, $\mathbf{X} \in \mathbb{R}^{N^+ \times F}$ denotes the node-feature matrix for $\mathcal{V}^+$.

Attributed nodes with partially known features mean that the node-feature matrix $\mathbf{X} \in \mathbb{R}^{N^+ \times F}$ has missing elements. $\mathcal{V}_u^{(d)}$ denotes a set of nodes with *unknown* features in $d$-th channel, where the nodes correspond to the rows with missing elements in the $d$-th column of $\mathbf{X}$. The set of remaining nodes with *known* features in the $d$-th channel is denoted by $\mathcal{V}_k^{(d)}$. Thus $\mathcal{V}_u^{(d)}$ and $\mathcal{V}_k^{(d)}$ become a partition of $\mathcal{V}^+$.

Under the above setting, we tackle the problem of *learning on a heterogeneous graph with missing node features*. The problem is to learn a function that produces desired output $\mathbf{Y}$ of a given task (*e.g.*, semi-supervised node classification or link prediction) on $\mathcal{G}$ with partially known node features. To this end, we aim to design an imputation scheme for missing features in $\mathbf{X}$ of $\mathcal{V}^+$, which maximizes the performance of a downstream GNN for a given task.

## 4 PROPOSED METHOD

### 4.1 OVERVIEW OF HETGFD

Figure 2 provides a brief overview of the proposed scheme called heterogeneous graph feature diffusion (HetGFD). HetGFD consists of two diffusion stages: preliminary diffusion and relation-aware diffusion. For relation-aware diffusion, distinct importance for each edge type is required. Thus, the preliminary diffusion stage first assumes equal importance for all edge types and generates a pre-imputed matrix. This pre-imputed matrix is then used to determine the importance of each edge type. The importance of each edge type enables relation-aware diffusion, which produces the final imputed matrix.

### 4.2 PRELIMINARY DIFFUSION

Diffusion-based methods (Rossi et al., 2022; Um et al., 2023) developed for homogeneous graphs update unknown features and utilize the unknown features as pathways to spread known features simultaneously. However, in a heterogeneous graph with non-attributed nodes, the diffusion is blocked by the absence of features. To tackle this challenge, we define virtual features filled with zeros for every non-attributed node in $\mathcal{V}^-$, where the dimension of the virtual features is set to the same dimension of features in the attributed node (*i.e.*, $F$). These virtual features are then iteratively updated through the preliminary diffusion propagating known features, enabling the missing features at attributed nodes in $\mathcal{V}^+$ to be updated via diffusion across all nodes in both $\mathcal{V}^-$ and $\mathcal{V}^+$. While we assume a single attributed node type, HetGFD can handle not only $\mathcal{G}$ with a single attributed node type but also $\mathcal{G}$ with multiple node types with different attribute distributions (*e.g.*, different feature dimensions). In such cases, we can impute missing features in each node type by applying HetGFD independently.

In our design, we adopt a channel-wise formulation with $d$-th channel. For notational convenience, we reorder all nodes in $\mathcal{V}$ in the order of $\mathcal{V}_k^{(d)}$, $\mathcal{V}_u^{(d)}$, and $\mathcal{V}^-$. According to the order of the reordered nodes, the feature vector and the $r$-type adjacency matrix can be written as:

$$\mathbf{x}^{(d)} = \begin{bmatrix} \mathbf{x}_k^{(d)} \\ \mathbf{x}_u^{(d)} \\ \mathbf{x}_-^{(d)} \end{bmatrix}, \quad \mathbf{A}^{(d,r)} = \begin{bmatrix} \mathbf{A}_{kk}^{(d,r)} & \mathbf{A}_{ku}^{(d,r)} & \mathbf{A}_{k-}^{(d,r)} \\ \mathbf{A}_{uk}^{(d,r)} & \mathbf{A}_{uu}^{(d,r)} & \mathbf{A}_{u-}^{(d,r)} \\ \mathbf{A}_{-k}^{(d,r)} & \mathbf{A}_{-u}^{(d,r)} & \mathbf{A}_{--}^{(d,r)} \end{bmatrix}, \tag{1}$$

where $\mathbf{x}_k^{(d)}$, $\mathbf{x}_u^{(d)}$, and $\mathbf{x}_-^{(d)}$ are column vectors that denote $d$-th channel feature vectors concatenated for all nodes in $\mathcal{V}_k^{(d)}$, $\mathcal{V}_u^{(d)}$, and $\mathcal{V}_-^{(d)}$, respectively. Here, $\mathcal{V}_k^{(d)}$ is referred to as source nodes. $\mathbf{x}_-^{(d)}$ consists of virtual feature values. Similarly, $\mathbf{A}^{(d,r)} \in \mathbb{R}^{N \times N}$ is composed of nine sub-matrices related to $\mathcal{V}_k^{(d)}$, $\mathcal{V}_u^{(d)}$, and $\mathcal{V}^-$. $\mathcal{E}^{(d)}$ denotes the edge set of the reordered nodes for $d$-th channel.

The preliminary diffusion is performed by iterative propagation using a transition matrix for each channel. Since the preliminary diffusion does not account for the importance of each edge type, assigning equal weights to all nodes when constructing the transition matrix may seem a reasonable approach. However, this approach can result in biased diffusion, where propagation predominantly occurs along the majority edge type. To address this issue, we introduce a penalty for the majority edge type when defining $\mathbf{A}^{(d,r)} \in \mathbb{R}^{N \times N}$:

$$\bar{\mathbf{A}}_{i,j}^{(d,r)} = \begin{cases} |\mathcal{E}_r|^{-1} & \text{if } \mathbf{A}_{i,j}^{(d,r)} \neq 0 \\ 0 & \text{otherwise} \end{cases} \tag{2}$$

We then sum $\mathbf{A}^{(d,r)}$ over all edge types to construct $\mathbf{A}^{(d)}$, *i.e.*, $\bar{\mathbf{A}}^{(d)} = \sum_{r \in \mathcal{R}} \bar{\mathbf{A}}^{(d,r)}$.

Next, to ensure that each updated feature is generated as a weighted average of neighboring features, we normalize $\bar{\mathbf{A}}^{(d)}$ to compute a a row-stochastic matrix $\mathbf{M}^{(d)} \in \mathbb{R}^{N \times N}$. Formally, $\mathbf{M}^{(d)} = (\mathbf{D}^{(d)})^{-1}\bar{\mathbf{A}}^{(d)}$ where $\mathbf{D}_{ii}^{(d)} = \sum_j \bar{\mathbf{A}}_{i,j}^{(d)}$. To preserve the known features $\mathbf{x}_k^{(d)}$ during the preliminary diffusion, we replace the first $|\mathcal{V}_k^{(d)}|$ rows of $\mathbf{M}^{(d)}$ with $\tilde{\mathbf{M}}^{(d)} \in \mathbb{R}^{N \times N}$:

$$\tilde{\mathbf{M}}^{(d)} = \begin{bmatrix} \mathbf{I} & \mathbf{0}_{ku} & \mathbf{0}_{k-} \\ \mathbf{M}_{uk}^{(d)} & \mathbf{M}_{uu}^{(d)} & \mathbf{M}_{u-}^{(d)} \\ \mathbf{M}_{-k}^{(d)} & \mathbf{M}_{-u}^{(d)} & \mathbf{M}_{--}^{(d)} \end{bmatrix}, \tag{3}$$

where $\mathbf{I} \in \mathbb{R}^{|\mathcal{V}_k^{(d)}| \times |\mathcal{V}_k^{(d)}|}$ is an identity matrix, and $\mathbf{0}_{ku}$ and $\mathbf{0}_{k-}$ are zero matrices.

Using $\tilde{\mathbf{M}}^{(d)}$ as a transition matrix, the preliminary diffusion is defined by

$$\tilde{\mathbf{x}}^{(d)}(t) = \tilde{\mathbf{M}}^{(d)}\tilde{\mathbf{x}}^{(d)}(t-1), \ \ t = 1, \cdots, K;$$

$$\tilde{\mathbf{x}}^{(d)}(0) = \begin{bmatrix} \mathbf{x}_k^{(d)} \\ \mathbf{0}_u \\ \mathbf{0}_- \end{bmatrix}, \tag{4}$$

where $\tilde{\mathbf{x}}^{(d)}(t)$ denotes an imputed feature vector after $t$ propagation steps. Here, $\mathbf{0}_u$ and $\mathbf{0}_-$ are zero column vectors with a length of $|\mathcal{V}_u^{(d)}|$ and $N^-$, respectively. As $K \to \infty$, the recursion converges, and $\tilde{\mathbf{x}}^{(d)}(K)$ reaches a steady state. We prove the convergence in Appendix A. It is noteworthy that we initialize the values with zeros based on this proof. The proof shows that initial values for $\mathbf{x}_u^{(d)}$ and $\mathbf{x}_-^{(d)}$ do not affect the steady state. With a sufficiently large $K$, we approximate the steady state $\lim_{t \to \infty} \tilde{\mathbf{x}}^{(d)}(t)$ to $\tilde{\mathbf{x}}^{(d)}(K)$. Since the imputed feature $\tilde{\mathbf{x}}^{(d)}(K)$ has different ordering from the original one, we rearrange $\tilde{\mathbf{x}}^{(d)}(K)$ to $\bar{\mathbf{x}}^{(d)}(K)$ in the original order. We then obtain a pre-imputed feature matrix $\bar{\mathbf{X}} \in \mathbb{R}^{N \times F}$ by concatenating $\{\bar{\mathbf{x}}^{(d)}(K)\}_{d=1}^F$ along the channels.

### 4.3 EDGE-TYPE-WISE HOMOPHILY

With the introduction of virtual features, pre-imputed features can be obtained for all nodes, enabling the comparison of features between attributed and non-attributed nodes. Feature homophily is the working principle of diffusion-based imputation methods developed for homogeneous graphs. However, in heterogeneous graphs, different edge types contribute to feature homophily in varying degrees. For example, two papers written by an author may have more similar features compared to two randomly chosen papers submitted to the same conference. Therefore, we aim to enhance message passing along edge types that significantly contribute to feature homophily in the relation-aware diffusion process. To quantify the contribution of each edge type, we define a new criterion named *edge-type-wise homophily*, denoted by $\mathcal{H}$.

**Definition 1.** *Edge-type-wise homophily* of edge type $r$ for a given feature matrix $\mathbf{X}$ is defined by

$$\mathcal{H}(r) = \frac{|\mathcal{E}_r|^{-1} \sum_{(v_i, r, v_j) \in \mathcal{E}_r} sim(\mathbf{X}_{i,:}, \mathbf{X}_{j,:})}{\mathbb{E}_{v_n, v_m \in \mathcal{V}}[sim(\mathbf{X}_{n,:}, \mathbf{X}_{m,:})]}, \tag{5}$$

where $sim(\cdot, \cdot)$ denotes cosine similarity between two row vectors, and $\mathbf{X}_{z,:}$ denotes the $z$-th row vector of $\mathbf{X}$. $\mathbb{E}_{v_n, v_m \in \mathcal{V}}[sim(\mathbf{X}_{n,:}, \mathbf{X}_{m,:})]$ is calculated from randomly sampled node pairs in $\mathcal{V}$.

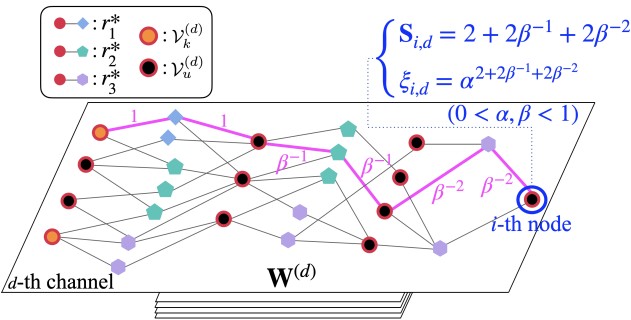

Figure 3: An illustrative example of relation-aware distance encoding. The shortest path distance from source nodes (SPD-S) is utilized to calculate pseudo-confidence (PC), which indicates the influence of each feature during diffusion. According to edge-type ranking based on $\mathcal{H}$, edge weights in $\boldsymbol{W}^{(d)}$ are assigned using $\beta$. The key idea is to rapidly decrease PC of features that are connected with observed features via low-ranking edges.

We calculate edge-type-wise homophily $\mathcal{H}(r)$ by using the pre-imputed feature matrix $\bar{\mathbf{X}} \in \mathbb{R}^{N \times F}$. Then, we rank the edge types in descending order of $\mathcal{H}(r)$, *i.e.*, $\mathcal{H}(r_1^*) > \ldots > \mathcal{H}(r_{|\mathcal{R}|}^*)$, which means that $r_k^*$ is the edge type with the $k$-th highest homophily. As a result, edge-type ranking is prepared prior to the relation-aware diffusion.

## 4.4 RELATION-AWARE DIFFUSION

To build a transition matrix for relation-aware diffusion, we utilize the concept of pseudo-confidence (PC) (Um et al., 2023). Pseudo-confidence (PC) represents the importance of each feature in the diffusion process. As PC of a feature increases, the feature can propagate its value more strongly. PC assumes that a missing feature located far from source nodes is more likely to be imputed with an inaccurate value. Consequently, Um et al. (2023) decreases PC as the distance between a feature and the nearest source nodes increases. PC is defined as:

**Definition 2.** *Pseudo-confidence of $\bar{\mathbf{X}}_{i,d}$ is defined by $\xi_{i,d} = \alpha^{\mathbf{S}_{i,d}}$ $(0 < \alpha < 1)$ where $\mathbf{S}_{i,d}$ denotes the shortest path distance between the $i$-th node and its nearest source node (SPD-S). $\alpha$ is a hyperparameter.*

Since PC is developed for homogeneous graphs, here $\mathbf{S}_{i,d} \in \{x \in \mathbb{Z} | x \geq 0\}$.

However, given the nature of heterogeneous graphs with multiple edge types, treating all edges equally can result in information loss. We enhance PC by incorporating edge-type-wise distinctions. We extend PC to weighted graphs, implying that $\mathbf{S}_{i,d} \in \{x \in \mathbb{R} | x \geq 0\}$ unlike existing PC. To obtain the weighted graphs where our new PC is calculated, we leverage the edge-type ranking.

Since PC and edge-type ranking are commonly based on feature homophily, the two concepts can be naturally combined. The key idea is to rapidly decrease the PC of a feature when it is connected to source nodes primarily through low-ranking edges (*i.e.*, edges belonging to a low-$\mathcal{H}$ edge type). Accordingly, we assign greater weights to low-ranking edges in the weighted graph $\mathbf{W}^{(d)}$ used for PC calculation.

Formally, a weighted adjacency matrix $\mathbf{W}^{(d,r)}$ to obtain $\mathbf{S}_{i,d}$ is defined as

$$\mathbf{W}_{i,j}^{(d,r)} = \begin{cases} \beta^{-(k-1)} & \text{if } \psi((v_i, r, v_j)) = r_k^* \\ 0 & \text{otherwise} \end{cases} \tag{6}$$

for $(v_i, r, v_j) \in \mathcal{E}^{(d)}$, where $\beta$ is a hyperparameter between 0 and 1 and $\psi(\cdot)$ is a numbering function from edge type $r$ to $r_a^*$ with a ranking $a$. We attain $\mathbf{W}^{(d)}$ by summing $\mathbf{W}^{(d,r)}$ for all edge type $r \in \mathcal{R}$.

Aligned with our key idea, as the ranking of edge type $r$ drops (*i.e.*, $\mathcal{H}(r)$ decreases or $k$ increases), the weights for $r$-type edges in $\mathbf{W}^{(d)}$ increase by dividing by $\beta$, as defined in Eq. (6). To calculate PC, defined as $\xi_{i,d} = \alpha^{\mathbf{S}_{i,d}}$, the shortest path distance from source nodes $\mathbf{S}_{i,d}$ on $\mathbf{W}^{(d)}$ is given by:

$$\mathbf{S}_{i,d} = \textit{SPD-S}(v_i | \mathcal{V}_k^{(d)}, \mathbf{W}^{(d)}), \tag{7}$$

where *SPD-S* is a function that outputs the shortest path distance from the $i$-th node to the source nodes (*i.e.*, the nodes in $\mathcal{V}_k^{(d)}$) on $\mathbf{W}^{(d)}$. By Definition 2, we can calculate PC of each feature value with $\mathbf{S}_{i,d}$ measured on $\mathbf{W}^{(d)}$. In Figure 3, the calculation process of PC is illustrated. It is important to note that $\mathbf{W}^{(d)}$ is not a transition matrix for relation-aware diffusion but is exclusively utilized for the calculation of PC.

In addition to PC for adjusting the importance of each feature during relation-aware diffusion, we assign greater weights to high-ranking edges (*i.e.*, edges belonging to a high-$\mathcal{H}$ edge type). To construct a transition matrix for relation-aware diffusion, we define an asymmetric weight matrix $\bar{\mathbf{W}}^{(d,r)}$ where element $\bar{\mathbf{W}}_{i,j}^{(d,r)}$ represents the weight of a directed edge from $j$-th node to $i$-th node, defined by:

$$\bar{\mathbf{W}}_{i,j}^{(d,r)} = \begin{cases} \beta^{k-1} \cdot \xi_{j,d}/\xi_{i,d} & \text{if } \psi\big((v_i, r, v_j)\big) = r_k^* \\ 0 & \text{otherwise} \end{cases} \tag{8}$$

for $(v_i, r, v_j) \in \mathcal{E}^{(d)}$. The term $\beta^{k-1}$ in Eq. (8) accomplishes an objective that the higher the ranking of an edge becomes ($k$ decreases), the higher the edge's weight becomes. The term $\xi_{j,d}/\xi_{i,d}$ strengthens the message passing from high-PC features to low-PC features, indicating that high-PC features propagate their values more strongly than low-PC features. If $\xi_{i,d}$ is lower than $\xi_{j,d}$, the term $\xi_{j,d}/\xi_{i,d}$ is larger than 1, which strengthens the weight of a directed edge from the $j$-th node to the $i$-th node.

We compute $\bar{\mathbf{W}}^{(d)}$ as $\bar{\mathbf{W}}^{(d)} = \sum_{r \in \mathcal{R}} \bar{\mathbf{W}}^{(d,r)}$. Next, we normalize $\bar{\mathbf{W}}^{(d)}$ to obtain a row-stochastic transition matrix $\mathbf{T}^{(d)}$, defined as $\mathbf{T}^{(d)} = (\mathbf{D}'^{(d)})^{-1}\bar{\mathbf{W}}^{(d)}$, where $\mathbf{D}'^{(d)}_{ii} = \sum_j \bar{\mathbf{W}}_{i,j}^{(d)}$. To preserve the known features, we replace the first $|\mathcal{V}_k^{(d)}|$ rows of $\mathbf{T}^{(d)}$ with $\widehat{\mathbf{T}}^{(d)}$ as follows:

$$\widehat{\mathbf{T}}^{(d)} = \begin{bmatrix} \mathbf{I} & \mathbf{0}_{ku} & \mathbf{0}_{k-} \\ \mathbf{T}_{uk}^{(d)} & \mathbf{T}_{uu}^{(d)} & \mathbf{T}_{u-}^{(d)} \\ \mathbf{T}_{-k}^{(d)} & \mathbf{T}_{-u}^{(d)} & \mathbf{T}_{--}^{(d)} \end{bmatrix}. \tag{9}$$

The relation-aware diffusion for final imputation is defined by

$$\hat{\mathbf{x}}^{(d)}(t) = \widehat{\mathbf{T}}^{(d)}\hat{\mathbf{x}}^{(d)}(t-1), \;\; t = 1, \cdots, K;$$
$$\hat{\mathbf{x}}^{(d)}(0) = \begin{bmatrix} \mathbf{x}_k^{(d)} \\ \mathbf{0}_u \\ \mathbf{0}_- \end{bmatrix}, \tag{10}$$

where $\hat{\mathbf{x}}^{(d)}(t)$ is an imputed feature vector in relation-aware diffusion through $t$ propagation steps. We use $\hat{\mathbf{x}}^{(d)}(K)$ with large enough $K$ to approximate a steady state of $\hat{\mathbf{x}}^{(d)}(t)$. After obtaining $\{\hat{\mathbf{x}}^{(d)}(K)\}_{d=1}^F$ via the channel-wise diffusion, we rearrange $\{\hat{\mathbf{x}}^{(d)}(K)\}_{d=1}^F$ in the original order. Then, we attain an imputed feature matrix $\hat{\mathbf{X}}$ by stacking the originally ordered vectors in $\{\hat{\mathbf{x}}^{(d)}(K)\}_{d=1}^F$ along the channels. Finally, we obtain $\hat{\mathbf{X}}' \in \mathbb{R}^{N^+ \times F}$ by removing all rows corresponding to $\mathcal{V}^-$ from $\hat{\mathbf{X}}$. $\hat{\mathbf{X}}'$ is an output of HetGFD and is fed to GNNs for downstream tasks.

In summary, our HetGFD framework addresses two critical challenges in heterogeneous graphs: handling non-attributed nodes through virtual features and accounting for distinct relations via edge-type-wise homophily $\mathcal{H}$. Leveraging $\mathcal{H}$, which aligns with the principles of diffusion-based imputation, HetGFD controls the overall diffusion process at both the feature level and the edge level in heterogeneous graphs. This dual control, with consideration of edge types, creates a significant gap compared to PCFI, which simply utilizes PC computed on unweighted graphs. This enables the use of information inherent in the edge types of heterogeneous graphs, resulting in imputed features that are beneficial for downstream tasks in such graphs.

## 5 EXPERIMENTS

### 5.1 EXPERIMENTAL SETUP

**Data Setting.** We conduct experiments on three widely used heterogeneous graph datasets (ACM, DBLP, and IMDB) (Jin et al., 2021) from different domains. Detailed descriptions of these datasets and their sources can be found in Appendix B.2. To conduct experiments on graphs with missing

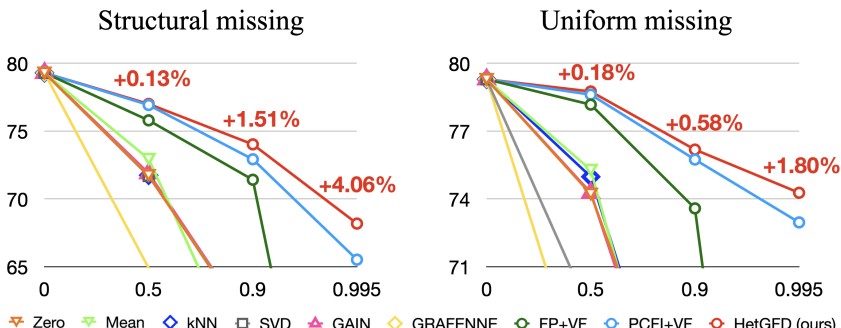

Figure 4: Performance averaged across the three datasets in semi-supervised node classification, measured in terms of Macro-F1 (%) under structural-missing settings with varying missing rates $r_m \in \{0, 0.5, 0.9, 0.995\}$. HGNN-AC is consistently used as a downstream GNN. VF denotes our virtual feature scheme. Figures highlighted in red indicate performance improvements over the most competitive baseline across each setting.

features, we remove a fixed rate of features in the datasets. Note that in the original data, raw features exist only for paper-type nodes in ACM and DBLP, and for movie-type nodes in IMDB. That is, only nodes of a single specific type (attributed-node type) are given raw features that represent semantic information. To compare the performance of imputation methods, we introduce missing values into a feature matrix from an attributed node type. Specifically, we remove features according to a missing rate, denoted as $r_m$, ranging from 0 to 1. In the positions where features are removed, we fill them with the value 'NaN'. We remove features in two ways as suggested in Um et al. (2023): (1) *structural missing:* After randomly selecting nodes with a ratio of $r_m$ among $t^+$-type nodes, we remove whole features from the selected nodes. That is, features in a node are either entirely known (observed) or entirely unknown (missing). (2) *uniform missing:* After randomly selecting feature values with a ratio of $r_m$ from the feature matrix $\mathbf{X}$, we remove the selected features. That is, we remove features uniformly across all $t^+$-type nodes.

Given a missing rate $r_m$, we randomly create 10 different binary masks of a feature matrix for each dataset. These masks indicate the location where feature values are missing, with a rate of $r_m$. We evaluate imputation methods across 10 runs, using the generated masks. To non-attributed nodes, we assign one-hot node features for each node type (*e.g.*, author and subject in ACM) so that a fair comparison of imputation methods for attributed nodes is enabled.

**Baselines.** We compare the performance of HetGFD with the following seven baselines: zero imputation, mean imputation, k-Nearest Neighbors (kNN) imputation (Troyanskaya et al., 2001), Iterative SVD (Troyanskaya et al., 2001), GAIN (Yoon et al., 2018), GRAFENNE (Gupta et al., 2023), FP+virtual features (FP+VF), PCFI (Um et al., 2023)+virtual features (PCFI+VF). It is noteworthy that our virtual feature scheme enables the use of existing diffusion-based methods, including FP and PCFI, on heterogeneous graphs.

Existing methods for attribute completion on heterogeneous graphs (He et al., 2022; Jin et al., 2021; Wang et al., 2022) cannot be compared since all these methods focus on learning representations for non-attributed nodes (*i.e.*, $\mathcal{V}^-$) under the assumption of no missing features in $t^+$-type nodes. In real-world situations, as there is no guarantee that missing data occurs only in specific types, our setting can be widely applied to various real-world applications. We demonstrate that HetGFD is in a collaborative relationship with attribute completion methods by utilizing HGNN-AC (Jin et al., 2021) as a downstream GNN.

**Implementation Details.** Except GRAFFENE which is a GNN framework, we feed a complete feature matrix obtained by the imputation methods into downstream GNNs. For a fair comparison of imputation methods, we adopt HGT (Hu et al., 2020b) for downstream GNNs, which is a state-of-the-art heterogeneous graph neural network. For link prediction, we utilize embeddings obtained from a GNN with HGT layers. To predict a link between a pair of nodes, we adopt the dot product of embeddings of the two nodes, as in Kipf & Welling (2016b); Lv et al. (2021). We further conduct experiments on semi-supervised node classification using HGNN-AC as a downstream GNN for the following two reasons: 1) to demonstrate the superiority of HetGFD across various GNNs; 2) to demonstrate that HetGFD can be combined with existing methods for attribute com-

Table 1: Semi-supervised node classification results (%) with $r_m = 0.995$.

| GNN | Missing type | Method | ACM | | DBLP | | IMDB | |
|---|---|---|---|---|---|---|---|---|
| | | | Macro-F1 | Micro-F1 | Macro-F1 | Micro-F1 | Macro-F1 | Micro-F1 |
| | | Full features | $92.52 \pm 0.41$ | $92.48 \pm 0.41$ | $91.28 \pm 1.11$ | $91.98 \pm 1.11$ | $54.51 \pm 4.74$ | $56.38 \pm 4.74$ |
| HGT | Structural missing | Zero | $82.75 \pm 1.40$ | $83.44 \pm 1.21$ | $90.11 \pm 1.46$ | $90.90 \pm 1.33$ | $44.29 \pm 1.83$ | $47.91 \pm 0.89$ |
| | | Mean | $83.25 \pm 1.20$ | $83.90 \pm 1.20$ | $89.88 \pm 2.02$ | $90.74 \pm 2.02$ | $45.38 \pm 2.21$ | $48.21 \pm 2.21$ |
| | | kNN | $82.67 \pm 1.54$ | $83.56 \pm 1.32$ | $90.37 \pm 1.67$ | $91.04 \pm 1.59$ | $44.88 \pm 3.25$ | $48.22 \pm 1.53$ |
| | | SVD | $82.71 \pm 1.37$ | $83.20 \pm 1.15$ | $90.11 \pm 1.46$ | $90.90 \pm 1.46$ | $44.28 \pm 1.82$ | $47.91 \pm 1.82$ |
| | | GAIN | $82.84 \pm 1.50$ | $83.58 \pm 1.27$ | $90.08 \pm 1.25$ | $90.85 \pm 1.11$ | $44.28 \pm 1.82$ | $47.91 \pm 0.89$ |
| | | GRAFENNE | $52.77 \pm 7.28$ | $65.49 \pm 5.17$ | $18.65 \pm 3.13$ | $27.73 \pm 0.94$ | $27.20 \pm 2.73$ | $36.12 \pm 1.76$ |
| | | FP+VF | $83.15 \pm 0.91$ | $83.60 \pm 0.79$ | $90.30 \pm 1.25$ | $91.07 \pm 1.14$ | $45.37 \pm 3.61$ | $48.30 \pm 0.98$ |
| | | PCFI+VF | $84.47 \pm 1.22$ | $85.06 \pm 0.98$ | $90.41 \pm 1.23$ | $91.14 \pm 1.08$ | $46.64 \pm 1.51$ | $49.69 \pm 1.83$ |
| | | HetGFD (ours) | $\mathbf{85.87 \pm 0.79}$ | $\mathbf{86.10 \pm 0.81}$ | $\mathbf{90.88 \pm 0.79}$ | $\mathbf{91.53 \pm 0.79}$ | $\mathbf{46.88 \pm 2.13}$ | $\mathbf{49.81 \pm 0.83}$ |
| | Uniform missing | Zero | $81.85 \pm 1.81$ | $82.67 \pm 1.59$ | $90.19 \pm 1.47$ | $90.99 \pm 1.33$ | $46.54 \pm 1.59$ | $48.49 \pm 0.98$ |
| | | Mean | $82.76 \pm 1.31$ | $83.48 \pm 1.31$ | $90.54 \pm 1.62$ | $91.32 \pm 1.62$ | $46.33 \pm 1.49$ | $48.32 \pm 1.49$ |
| | | kNN | $81.91 \pm 0.85$ | $82.43 \pm 0.39$ | $90.50 \pm 1.69$ | $91.20 \pm 1.56$ | $45.81 \pm 2.30$ | $48.70 \pm 1.15$ |
| | | SVD | $80.45 \pm 2.22$ | $81.52 \pm 2.17$ | $90.33 \pm 1.46$ | $91.07 \pm 1.32$ | $44.53 \pm 6.42$ | $48.38 \pm 1.26$ |
| | | GAIN | $82.98 \pm 0.78$ | $83.84 \pm 0.78$ | $90.02 \pm 1.41$ | $91.01 \pm 1.41$ | $46.54 \pm 1.59$ | $48.49 \pm 1.59$ |
| | | GRAFENNE | $73.04 \pm 1.34$ | $74.29 \pm 1.06$ | $17.80 \pm 2.24$ | $28.61 \pm 0.91$ | $36.51 \pm 1.88$ | $39.82 \pm 1.60$ |
| | | FP+VF | $83.89 \pm 1.08$ | $84.49 \pm 1.08$ | $90.15 \pm 1.49$ | $90.91 \pm 1.49$ | $45.58 \pm 2.01$ | $48.61 \pm 2.01$ |
| | | PCFI+VF | $86.26 \pm 0.69$ | $86.61 \pm 0.63$ | $90.49 \pm 1.43$ | $91.14 \pm 1.35$ | $47.25 \pm 1.88$ | $49.94 \pm 1.38$ |
| | | HetGFD (ours) | $\mathbf{88.14 \pm 0.69}$ | $\mathbf{88.14 \pm 0.66}$ | $\mathbf{90.65 \pm 1.53}$ | $\mathbf{91.40 \pm 1.24}$ | $\mathbf{48.57 \pm 1.41}$ | $\mathbf{50.57 \pm 1.66}$ |

Table 2: Link prediction results (%) with $r_m = 0.995$. AUC denotes the ROC AUC score.

| Missing type | Method | ACM | | DBLP | | IMDB | |
|---|---|---|---|---|---|---|---|
| | | AUC | AP | AUC | AP | AUC | AP |
| | Full features | $76.25 \pm 1.20$ | $77.64 \pm 1.07$ | $71.52 \pm 0.51$ | $66.92 \pm 0.66$ | $92.47 \pm 1.06$ | $86.94 \pm 1.61$ |
| Structural missing | Zero | $71.65 \pm 2.16$ | $71.74 \pm 3.55$ | $72.49 \pm 0.63$ | $74.21 \pm 0.60$ | $92.48 \pm 1.06$ | $86.95 \pm 1.60$ |
| | Mean | $71.64 \pm 1.30$ | $71.66 \pm 1.33$ | $72.49 \pm 0.63$ | $74.20 \pm 0.60$ | $91.78 \pm 1.13$ | $85.80 \pm 2.12$ |
| | kNN | $72.04 \pm 1.66$ | $72.55 \pm 2.11$ | $71.96 \pm 1.37$ | $69.86 \pm 1.89$ | $91.10 \pm 1.07$ | $84.44 \pm 1.97$ |
| | SVD | $71.49 \pm 1.77$ | $72.29 \pm 2.13$ | $72.49 \pm 0.63$ | $74.21 \pm 0.60$ | $92.48 \pm 1.06$ | $86.95 \pm 1.60$ |
| | GAIN | $72.22 \pm 1.19$ | $73.21 \pm 1.10$ | $72.49 \pm 0.63$ | $74.20 \pm 0.61$ | $92.48 \pm 1.06$ | $86.95 \pm 1.60$ |
| | GRAFENNE | $74.87 \pm 6.71$ | $67.60 \pm 5.87$ | $90.14 \pm 7.26$ | $76.53 \pm 7.12$ | $82.38 \pm 5.75$ | $69.72 \pm 4.60$ |
| | FP+VF | $73.40 \pm 0.75$ | $74.03 \pm 0.84$ | $71.58 \pm 0.85$ | $70.01 \pm 1.43$ | $\mathbf{92.50 \pm 1.04}$ | $\mathbf{86.99 \pm 1.58}$ |
| | PCFI+VF | $73.41 \pm 1.16$ | $73.22 \pm 1.18$ | $71.37 \pm 0.55$ | $66.78 \pm 0.74$ | $91.71 \pm 1.33$ | $85.37 \pm 2.08$ |
| | HetGFD (ours) | $\mathbf{78.25 \pm 1.34}$ | $\mathbf{78.62 \pm 2.12}$ | $\mathbf{91.94 \pm 0.67}$ | $\mathbf{91.88 \pm 0.91}$ | $\mathbf{92.50 \pm 1.04}$ | $\mathbf{86.99 \pm 1.58}$ |
| Uniform missing | Zero | $70.69 \pm 1.48$ | $70.17 \pm 3.07$ | $72.48 \pm 0.62$ | $74.20 \pm 0.60$ | $\mathbf{92.50 \pm 1.04}$ | $\mathbf{86.99 \pm 1.58}$ |
| | Mean | $71.98 \pm 1.02$ | $72.02 \pm 0.96$ | $72.48 \pm 0.62$ | $74.20 \pm 0.60$ | $91.40 \pm 1.14$ | $85.33 \pm 1.93$ |
| | kNN | $71.02 \pm 1.49$ | $72.49 \pm 2.46$ | $72.72 \pm 1.85$ | $70.29 \pm 3.73$ | $91.15 \pm 1.09$ | $84.50 \pm 2.04$ |
| | SVD | $70.49 \pm 2.11$ | $70.70 \pm 4.08$ | $72.48 \pm 0.62$ | $74.20 \pm 0.60$ | $\mathbf{92.50 \pm 1.04}$ | $\mathbf{86.99 \pm 1.58}$ |
| | GAIN | $71.92 \pm 0.92$ | $73.17 \pm 1.09$ | $72.48 \pm 0.62$ | $74.20 \pm 0.60$ | $\mathbf{92.50 \pm 1.04}$ | $\mathbf{86.99 \pm 1.58}$ |
| | GRAFENNE | $74.76 \pm 9.82$ | $72.96 \pm 9.71$ | $63.78 \pm 31.28$ | $61.86 \pm 28.14$ | $80.69 \pm 15.81$ | $73.22 \pm 14.33$ |
| | FP+VF | $73.18 \pm 0.96$ | $73.77 \pm 0.82$ | $71.86 \pm 1.66$ | $70.03 \pm 1.97$ | $91.52 \pm 1.15$ | $85.67 \pm 2.14$ |
| | PCFI+VF | $74.94 \pm 1.37$ | $73.80 \pm 1.63$ | $70.76 \pm 3.14$ | $68.97 \pm 3.85$ | $91.54 \pm 1.13$ | $85.70 \pm 2.08$ |
| | HetGFD (ours) | $\mathbf{76.96 \pm 1.74}$ | $\mathbf{77.19 \pm 1.98}$ | $\mathbf{92.17 \pm 0.56}$ | $\mathbf{92.12 \pm 0.53}$ | $91.95 \pm 1.72$ | $86.72 \pm 3.40$ |

pletion in heterogeneous graphs, which assume full features in $t^+$-type nodes. We include further details on experiments (*e.g.*, hyperparameter tuning, baseline implementation, train/validation/test splits, and training details) in Appendix B. Further experimental results, including ablation study, time complexity, and qualitative results, can be found in Appendix D. Due to space limitations, the experimental setup for the PPI dataset (Zitnik & Leskovec, 2017) in the biomedical domain is described in Appendix B.5.

## 5.2 SEMI-SUPERVISED NODE CLASSIFICATION RESULTS

Figure 4 shows the trend of Macro-F1 scores for the compared methods when HGNN-AC models are used for downstream GNNs across all the methods. Macro-F1 tends to decrease across all methods as the missing rate $r_m$ increases. Notably, the compared methods, except for the diffusion-based methods, exhibit significant performance degradation. This result suggests that diffusion-based methods are also effective in heterogeneous graphs. Among the diffusion-based methods, Het-GFD outperforms both FP+VF and PCFI+VF. Moreover, the performance gain of HetGFD tends to increase as $r_m$ increases. This is because when $r_m$ is low, the other methods can also maintain a certain level of performance using remaining known features. However, when $r_m$ increases, the importance of imputation becomes greater as the proportion of imputed values in an output matrix increases. HetGFD demonstrates remarkable resistance to high $r_m$, as evidenced by the results obtained from different settings. Table 1 shows the overall results of semi-supervised node classification with $r_m = 0.995$ when HGT is used as the downstream GNN (results for HGNN-AC are provided in Appendix F). HetGFD shows superior performance compared to all other methods regardless of a downstream GNN in all settings with $r_m = 0.995$.

## 5.3 LINK PREDICTION RESULTS

The link prediction results under the missing settings with $r_m = 0.995$ are shown in Table 2. During the experiments, we observe that Zero, Mean, SVD, and GAIN impute values very close to 0 for the missing features at $r_m = 0.995$. We found that this observation leads to similar performance

Table 3: Performance on the protein-protein interaction networks (PPI) dataset for different $r_m$, measured by accuracy (%). VF denotes our virtual feature scheme. 'Impr.' indicates performance improvements over the most competitive baseline at each $r_m$.

| $r_m$ | 0 | 0.5 | 0.9 | 0.995 |
|---|---|---|---|---|
| Zero | $98.49 \pm 0.13$ | $78.74 \pm 1.01$ | $64.15 \pm 1.18$ | $62.20 \pm 0.24$ |
| Mean | $98.49 \pm 0.13$ | $64.40 \pm 1.97$ | $64.40 \pm 1.97$ | $62.14 \pm 0.15$ |
| kNN | $98.49 \pm 0.13$ | $78.74 \pm 1.01$ | $64.15 \pm 1.18$ | $62.20 \pm 0.24$ |
| SVD | $98.49 \pm 0.13$ | $79.30 \pm 1.15$ | $64.10 \pm 1.23$ | $62.23 \pm 0.26$ |
| GAIN | $98.49 \pm 0.13$ | $78.85 \pm 1.09$ | $64.13 \pm 1.09$ | $62.20 \pm 0.24$ |
| GRAFENNE | $83.76 \pm 9.15$ | $63.97 \pm 1.87$ | $63.15 \pm 1.31$ | $62.26 \pm 0.00$ |
| FP+VF | $98.49 \pm 0.13$ | $80.44 \pm 2.34$ | $64.78 \pm 1.51$ | $62.20 \pm 0.24$ |
| PCFI+VF | $98.49 \pm 0.13$ | $80.75 \pm 1.68$ | $65.22 \pm 2.19$ | $62.14 \pm 0.32$ |
| HetGFD (ours) | $98.49 \pm 0.13$ | $\mathbf{81.57 \pm 1.04}$ | $\mathbf{66.84 \pm 1.92}$ | $\mathbf{63.20 \pm 0.37}$ |
| Impr. | - | $+1.02\%$ | $+2.48\%$ | $+1.51\%$ |

among these four methods. On the IMDB dataset, the results show that the overall methods perform similarly to when full features are used. This indicates that the information contained in the features is not useful for link prediction on IMDB. HetGFD achieves state-of-the-art performance across all settings except for IMDB with uniform missing. Features obtained by HetGFD contain rich structural information, taking edge types into consideration, which leads to performance improvement in heterogeneous graphs.

## 5.4 APPLICABILITY TO THE BIOMEDICAL DOMAIN

To demonstrate the applicability of HetGFD to biomedical domain, we conduct an experiment on the protein-protein interaction networks (PPI) dataset, which is used for analyzing biological relevance and disease understanding. Table 3 shows the semi-supervised node classification performance of different imputation methods on the PPI dataset, where structural missing is applied using different $r_m$. As shown in the table, the accuracy of all methods commonly decreases as $r_m$ increases. However, our HetGFD consistently shows the best performance across various $r_m$. Among type A and type B edge types, we observe that type A exhibits higher edge-type-wise homophily, and HetGFD assigns more importance to type A edges. We attribute the performance gains of HetGFD to the correlation between the edge types and the class label. Unlike existing methods, HetGFD differentiates between edge types, enabling it to more effectively utilize the information contained within them.

## 6 COMPLEXITY ANALYSIS

The time complexity of HetGFD involves three main processes: two diffusion stages ($O(|\mathcal{E}|)$), edge-type-wise homophily calculation ($O(F \cdot |\mathcal{E}|)$), and PC calculation ($O(N^2)$). In structural-missing settings, a single PC calculation is needed ($O(F \cdot |\mathcal{E}| + N^2)$), while in uniform-missing settings, $F$ PC calculations are required ($O(F \cdot |\mathcal{E}| + F \cdot N^2)$). Notably, PCFI, the most competitive baseline, also involves PC calculation with $O(N^2)$. Compared to PCFI, HetGFD consistently demonstrates superior performance across all experimental settings. A more detailed time complexity analysis is provided in Appendix D.3.

## 7 CONCLUSION

In this paper, we introduce virtual features that enable the use of diffusion-based imputation on heterogeneous graphs. Building on these virtual features, we propose a novel imputation method called Heterogeneous Graph Feature Diffusion (HetGFD) tailored to heterogeneous graphs. HetGFD ranks each edge type according to edge-type-wise homophily, and this edge-type ranking facilitates relation-aware distance encoding. By treating each edge type differently, HetGFD, using relation-aware diffusion, shows its superiority over state-of-the-art methods on semi-supervised node classification and link prediction tasks across various domains. We further confirm that our virtual feature scheme effectively transfers the advantages of existing diffusion-based methods to the heterogeneous graph domain. We believe that our work will significantly contribute to solving missing data problems in various real-world scenarios that contain heterogeneity, due to the effectiveness and rapid imputation time of HetGFD. However, its effectiveness may be limited on feature-heterophilic heterogeneous graphs, where most paths connecting attributed nodes are heterophilic connections. Additionally, HetGFD will be effective when all nodes belonging to a node type with features have features of the same nature and scale.

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

## A  CONVERGENCE PROOF FOR DIFFUSION IN HETGFD

HetGFD is comprised of two diffusion stages: preliminary diffusion and relation-aware diffusion. We prove the convergence of the two diffusion stages as follows.

**Proposition 1.** *A row-stochastic transition matrix $\tilde{\boldsymbol{M}}^{(d)}$ for the preliminary diffusion is expressed as*

$$\tilde{\boldsymbol{M}}^{(d)} = \begin{bmatrix} \boldsymbol{I} & \boldsymbol{0}_{ku} & \boldsymbol{0}_{k-} \\ \boldsymbol{M}_{uk}^{(d)} & \boldsymbol{M}_{uu}^{(d)} & \boldsymbol{M}_{u-}^{(d)} \\ \boldsymbol{M}_{-k}^{(d)} & \boldsymbol{M}_{-u}^{(d)} & \boldsymbol{M}_{--}^{(d)} \end{bmatrix}.$$

*The recursive formula of preliminary diffusion is defined by*

$$\tilde{\boldsymbol{x}}^{(d)}(t) = \begin{bmatrix} \tilde{\boldsymbol{x}}_k^{(d)}(t) \\ \tilde{\boldsymbol{x}}_u^{(d)}(t) \\ \tilde{\boldsymbol{x}}_-^{(d)}(t) \end{bmatrix} = \tilde{\boldsymbol{M}}^{(d)} \tilde{\boldsymbol{x}}^{(d)}(t-1), \;\; t = 1, \cdots, K;$$

$$\tilde{\boldsymbol{x}}^{(d)}(0) = \begin{bmatrix} \boldsymbol{x}_k^{(d)} \\ \boldsymbol{0}_u \\ \boldsymbol{0}_- \end{bmatrix}.$$

*Then, $\lim\limits_{K \to \infty} \tilde{\boldsymbol{x}}_u^{(d)}(K)$ converges.*

The proof of Proposition 1 follows proofs in Rossi et al. (2022); Um et al. (2023). Um et al. (2023) proves convergence of feature diffusion with a row-stochastic transition matrix. Here, we prove the case when non-attributed nodes participate in feature diffusion with virtual features. We begin with two lemmas.

**Lemma 1.** *$\boldsymbol{M}^{(d)}$ is a row-stochastic matrix obtained by $\boldsymbol{M}^{(d)} = (\boldsymbol{D}^{(d)})^{-1} \boldsymbol{M}^{(d)}$ where $\boldsymbol{M}^{(d)}$ is a weighted-adjacency matrix of a connected graph $\mathcal{G}$ and $\boldsymbol{D}_{ii}^{(d)} = \sum_j \boldsymbol{M}_{i,j}^{(d)}$. Let $\boldsymbol{M}_{nn}^{(d)} \in \mathbb{R}^{(N-|\mathcal{V}_u^{(d)}|) \times (N-|\mathcal{V}_u^{(d)}|)}$ be the bottom-right submatrix of $\boldsymbol{M}^{(d)}$ and $\rho(\cdot)$ be spectral radius. Then, $\rho(\boldsymbol{M}_{nn}^{(d)}) < 1$.*

*Proof.* Let $\boldsymbol{M}_{nn0}^{(d)} \in \mathbb{R}^{N \times N}$ be a matrix defined by

$$\boldsymbol{M}_{nn0}^{(d)} = \begin{bmatrix} \boldsymbol{0}_{kk} & \boldsymbol{0}_{kn} \\ \boldsymbol{0}_{nk} & \boldsymbol{M}_{nn}^{(d)} \end{bmatrix},$$

where $\boldsymbol{0}_{kk} \in \{0\}^{|\mathcal{V}_k^{(d)}| \times |\mathcal{V}_k^{(d)}|}$, $\boldsymbol{0}_{kn} \in \{0\}^{|\mathcal{V}_k^{(d)}| \times (N-|\mathcal{V}_k^{(d)}|)}$, and $\boldsymbol{0}_{nk} \in \{0\}^{(N-|\mathcal{V}_k^{(d)}|) \times |\mathcal{V}_k^{(d)}|}$. Since $\boldsymbol{M}^{(d)}$ is the weighted adjacency matrix of the connected graph $\mathcal{G}$, $\boldsymbol{M}_{nn0}^{(d)} \leq \boldsymbol{M}^{(d)}$ element-wisely and $\boldsymbol{M}_{nn0}^{(d)} \neq \boldsymbol{M}$. Furthermore, $\boldsymbol{M}_{nn0}^{(d)} + \boldsymbol{M}^{(d)}$ is an adjacency matrix of a connected graph. By Theorem 2.2.7 of Berman & Plemmons (1994), $\boldsymbol{M}_{nn0}^{(d)} + \boldsymbol{M}^{(d)}$ is irreducible. By Corollary 2.1.5 of Berman & Plemmons (1994), $\rho(\boldsymbol{M}_{nn0}^{(d)}) < \rho(\boldsymbol{M}^{(d)})$ and by Theorem 2.5.3, $\rho(\boldsymbol{M}^{(d)}) = 1$. Then, $\rho(\boldsymbol{M}_{nn0}^{(d)}) = \rho(\boldsymbol{M}_{nn}^{(d)})$ since $\boldsymbol{M}_{nn0}^{(d)}$ and $\boldsymbol{M}_{nn}^{(d)}$ have the same non-zero eigenvalues. Therefore, $\rho(\boldsymbol{M}_{nn0}^{(d)}) = \rho(\boldsymbol{M}_{nn}^{(d)}) < 1$. $\qquad\square$

**Lemma 2.** *$\boldsymbol{I} - \boldsymbol{M}_{nn}^{(d)}$ is invertible where $\boldsymbol{I}$ is the $(N - |\mathcal{V}_k^{(d)}|) \times (N - |\mathcal{V}_k^{(d)}|)$ identity matrix.*

*Proof.* By Lemma. 1 1 is not an eigenvalue of $\boldsymbol{M}_{nn}^{(d)}$. Therefore, 0 is not an eigenvalue of $\boldsymbol{I} - \boldsymbol{M}_{nn}^{(d)}$. Hence $\boldsymbol{I} - \boldsymbol{M}_{nn}^{(d)}$ is invertible. $\qquad\square$

We now prove Proposition 1 as follows.

*Proof.* Let $\boldsymbol{M}_{nk}^{(d)} \in \mathbb{R}^{(N-|\mathcal{V}_k^{(d)}|) \times |\mathcal{V}_k^{(d)}|}$ and $\boldsymbol{M}_{nn}^{(d)} \in \mathbb{R}^{(N-|\mathcal{V}_k^{(d)}|) \times (N-|\mathcal{V}_k^{(d)}|)}$ be the bottom-left and bottom-right submatrices of $\boldsymbol{M}^{(d)}$, respectively. Let $\tilde{\boldsymbol{x}}_n^{(d)}(t) = \begin{bmatrix} \tilde{\boldsymbol{x}}_u^{(d)}(t) \\ \tilde{\boldsymbol{x}}_-^{(d)}(t) \end{bmatrix}$. Then, we can unfold the

recursive formula of preliminary diffusion as

$$\tilde{\boldsymbol{x}}^{(d)}(t) = \begin{bmatrix} \tilde{\boldsymbol{x}}_k^{(d)}(t) \\ \tilde{\boldsymbol{x}}_n^{(d)}(t) \end{bmatrix} = \tilde{\boldsymbol{M}}^{(d)}\tilde{\boldsymbol{x}}^{(d)}(t-1)$$

$$= \begin{bmatrix} \boldsymbol{I} & \boldsymbol{0}_{kn} \\ \boldsymbol{M}_{nk}^{(d)} & \boldsymbol{M}_{nn}^{(d)} \end{bmatrix} \begin{bmatrix} \tilde{\boldsymbol{x}}_k^{(d)}(t-1) \\ \tilde{\boldsymbol{x}}_n^{(d)}(t-1) \end{bmatrix}$$

$$= \begin{bmatrix} \tilde{\boldsymbol{x}}_k^{(d)}(t-1) \\ \boldsymbol{M}_{nk}^{(d)}\tilde{\boldsymbol{x}}_k^{(d)}(t-1) + \boldsymbol{M}_{nn}^{(d)}\tilde{\boldsymbol{x}}_n^{(d)}(t-1) \end{bmatrix}.$$

In $|\mathcal{V}_k^{(d)}|$ rows from the top, $\tilde{\boldsymbol{x}}_k^{(d)}(t)$ does not change from $\boldsymbol{x}_k^{(d)}$. Hence we only consider the remaining rows. We unroll the recursion in the remaining $N - |\mathcal{V}_k^{(d)}|$ rows as follows.

$$\tilde{\boldsymbol{x}}_n^{(d)}(K) = \boldsymbol{M}_{nk}^{(d)}\boldsymbol{x}_k^{(d)} + \boldsymbol{M}_{nn}^{(d)}\tilde{\boldsymbol{x}}_n^{(d)}(K-1)$$

$$= \dots$$

$$= \left(\sum_{t=0}^{K-1}(\boldsymbol{M}_{nn}^{(d)})^t\right)\boldsymbol{M}_{nk}^{(d)}\boldsymbol{x}_k^{(d)} + (\boldsymbol{M}_{nn}^{(d)})^K\tilde{\boldsymbol{x}}_n^{(d)}(0)$$

By Lemma 1 that $\lim_{K\to\infty}(\boldsymbol{M}_{nn}^{(d)})^K = 0$, $\lim_{K\to\infty}(\boldsymbol{M}_{nn}^{(d)})^K\tilde{\boldsymbol{x}}_n^{(d)}(0) = 0$ regardless of initial values for $\tilde{\boldsymbol{x}}_n^{(d)}(0)$ (We set the initial values to zeros in implementation for simplicity).

Since $\lim_{K\to\infty}\sum_{t=0}^{K-1}(\boldsymbol{M}_{nn}^{(d)})^t = (\boldsymbol{I} - \boldsymbol{M}_{nn}^{(d)})^{-1}$ and $\rho(\boldsymbol{M}_{nn}^{(d)}) < 1$ by Lemma. 1, $\boldsymbol{I} - \boldsymbol{M}_{nn}^{(d)}$ is invertible. Then,

$$\lim_{K\to\infty}\tilde{\boldsymbol{x}}_n^{(d)}(K) = \lim_{K\to\infty}\left(\sum_{t=0}^{K-1}(\boldsymbol{M}_{nn}^{(d)})^t\right)\boldsymbol{M}_{nk}^{(d)}\boldsymbol{x}_k^{(d)}$$

$$= (\boldsymbol{I} - \boldsymbol{M}_{nn}^{(d)})^{-1}\boldsymbol{M}_{nk}^{(d)}\boldsymbol{x}_k^{(d)}.$$

Here, $\lim_{K\to\infty}\tilde{\boldsymbol{x}}_u^{(d)}(K)$ is the first $|\mathcal{V}_k^{(d)}|$ rows of $\lim_{K\to\infty}\tilde{\boldsymbol{x}}_n^{(d)}(K)$ that converges. □

In the same way, the convergence of relation-aware diffusion can be proved by simply replacing $\tilde{\boldsymbol{M}}^{(d)}$ with $\hat{\boldsymbol{T}}^{(d)}$.

## B  FURTHER DETAILS ON IMPLEMENTATION

All the models used in this paper are implemented with Pytorch (Paszke et al., 2017) and Pytorch Geometric (Fey & Lenssen, 2019). All experiments are conducted with an Intel Core I5-6600 CPU @ 3.30 GHz and a single GPU (NVIDIA GeForce RTX 2080 Ti).

### B.1  BASELINES

We compare the performance of HetGFD with the following seven baselines. **(1) Zero imputation (Zero)** imputes all missing features with zero values. **(2) Mean imputation (Mean)** imputes a missing feature value at $\boldsymbol{X}_{i,j}$ with the mean of all known feature values in the $j$-th channel. **(3) Iterative SVD (SVD)** (Troyanskaya et al., 2001) is a structure-agnostic method that imputes missing features by performing iterative low-rank SVD decomposition for matrix completion. **(4) GAIN** (Yoon et al., 2018) is a structure-agnostic deep imputation method that employs generative adversarial training. **(5) GRAFENNE** (Gupta et al., 2023) is a GNN architecture that addresses heterogeneous features. **(6) FP** (Rossi et al., 2022)**+virtual features (FP+VF)**. FP is a diffusion-based imputation method designed for homogeneous graphs, where known features diffuse channel-wisely. To enable diffusion, we introduce our virtual feature scheme to FP. **(7) PCFI** (Um et al., 2023)**+virtual features**

Table 4: Dataset statistics.

| Datasets | Nodes | Edges | # classes | Feature dimension | Target node type | Target edge type |
|---|---|---|---|---|---|---|
| ACM | # Paper: 4014
# Author: 7157
# Subject: 56 | # Paper-Paper: 9612
# Paper-Author: 26794
# Paper-Subject: 8028 | 3 | 4000 | Paper | Paper-Author |
| DBLP | # Author: 4057
# Paper: 14328
# Term: 7723
# Conference: 20 | # Paper-Author: 39290
# Paper-Term: 171620
# Paper-Conference: 28656 | 4 | 4231 | Author | Paper-Author |
| IMDB | # Movie: 4025
# Director: 1836
# Actor: 4523 | # Movie-Director: 8050
# Movie-Actor: 24144 | 5 | 3066 | Movie | Movie-Director |

**(PCFI+VF)**. PCFI is a state-of-the-art diffusion-based method for homogeneous graphs, which utilizes pseudo-confidence of each feature value during the diffusion. Similarly, our virtual feature scheme is employed in PCFI as well.

For all baselines, we use their original hyperparameters and tuning methods specified in the respective papers and official codes. For two simple baselines, Zero imputation (Zero) and Mean imputation (Mean), we implement them with Pytorch built-in function. For the other baselines, we implement them as follows.

- k-nearest neighbor (kNN) imputation (Troyanskaya et al., 2001). We use a custom our implementation that performs imputation by creating $k$ edges for each node based on cosine similarity and leveraging the neighborhood mean. For each setting, $k$ is search within $\{1, 3, 5, 10\}$.

- Iterative SVD (SVD) (Troyanskaya et al., 2001). We use the implementation from the fancyimpute package.[1] We tune hyperparameter $rank$ in $\{F/5, F-1\}$ as in the implemented code from the package.

- Generative adversarial imputation nets (GAIN) (Yoon et al., 2018). We use the source code[2] released by the authors. We set all the hyperparameters of GAIN to those in the released source code.

- GRAFENNE (Gupta et al., 2023). We use the source code[3] released by the authors. We set all the hyperparameters of GRAFENNE to those in the released source code.

- Feature propagation (FP) (Rossi et al., 2022). We use the source code[4] released by the authors.

- Pseudo-Confidence-based feature imputation (PCFI) (Um et al., 2023). We use the source code[5] released by the authors. For PCFI models, we tune hyperparameter $\alpha, \beta$ in the search range specified in Um et al. (2023).

While the codes for SVD, FP, PCFI are Apache-2.0 licensed, the codes for GAIN and GRAFENNE have no public declaration of license.

## B.2 DATASETS

ACM is a citation network consisting of three node types and three edge types. DBLP is a bibliography network consisting of four node types and three edge types. IMDB, extracted from an online movie database, includes three node types and two edge types. We downloaded all the datasets used in this paper from the GitHub repository for Jin et al. (2021). This publicly available repository does not contain any statements ragarding licenses for the datasets. We conducted all the experiments on the largest connected components of each dataset. For a disconnected graph, HetGFD can deal with it by working on each connected component independently. The statistics of the three datasets are summarized in Table 4.

---

[1]https://github.com/iskandr/fancyimpute

[2]https://github.com/jsyoon0823/GAIN

[3]https://github.com/data-iitd/Grafenne

[4]https://github.com/twitter-research/feature-propagation

[5]https://github.com/daehoum1/pcfi

### B.3 SEMI-SUPERVISED NODE CLASSIFICATION

We utilize the node split suggested in Jin et al. (2021), which uses $10\%$ nodes for training, $10\%$ nodes for validation, and $80\%$ nodes for testing. When training HGT (Hu et al., 2020b) models, we use Adam optimizer (Kingma & Ba, 2014). The maximum number of epochs is set to $1000$ and we apply an early stopping strategy with the patience of $200$ epochs. We tune hyperparameters for training downstream GNN models and conduct a grid search based on the validation sets. Specifically, we search for the optimal number of layers from $\{1, 2, 3\}$ and the learning rate from $\{0.1, 0.01, 0.001, 0.0001\}$. We set the the hidden dimension to $64$ for all the models. In the experiments with HGNN-AC (Jin et al., 2021), we set a learning strategy and all the parameters of HGNN-AC models according to the official code[6] for Jin et al. (2021).

### B.4 LINK PREDICTION

For the link prediction splits, as described in Kipf & Welling (2016b), we divide target edges into training, validation, and testing sets, comprising $10\%$, $5\%$, and $85\%$ of the edges, respectively. In our approach, negative sampling is performed to generate non-existent edges for training, validation, and testing in link prediction tasks. First, we create a mask that identifies all potential edges in the graph, excluding existing edges to form a pool of non-edges. From this pool, we randomly sample negative edges, ensuring the number of negative samples matches the desired ratio for validation and testing. The remaining non-edges are used to create a training mask for negative samples. For each dataset, we generate 10 edge splits. We utilize the Adam optimizer (Kingma & Ba, 2014) to train the HGT models with the hidden dimension of $64$. Similar to HGT training in semi-supervised node classification, we tune the hyperparameters for training the downstream GNN models and perform a grid search based on the validation sets. Specifically, we search for the optimal number of layers from $\{1, 2, 3\}$ and the learning rate from $\{0.1, 0.01, 0.001, 0.0001\}$. It is important to note that both imputation and the training of downstream GNNs are performed exclusively on training edges in each split, with validation and testing edges excluded.

For evaluation, we utilize AUC and AP, two metrics widely used for link prediction tasks. The ROC AUC metric (denoted as AUC) evaluates the model's ability to distinguish between positive (true) links and negative (non-existent) links. It is computed by assessing the True Positive Rate (TPR) and False Positive Rate (FPR) at various thresholds of the predicted edge scores. These values are used to construct a Receiver Operating Characteristic (ROC) curve, which plots FPR on the x-axis and TPR on the y-axis. The Area Under the Curve (AUC) is then calculated, providing a single value that represents the overall performance of the model. The AP (Average Precision) metric emphasizes the precision-recall tradeoff, making it particularly valuable for imbalanced datasets. It is calculated by measuring Precision and Recall values across different thresholds of the predicted edge scores. These values are used to create a Precision-Recall curve, and the AP score is derived as the weighted average of precision values at each level of recall. A higher AP score reflects better performance, especially in identifying true links among a large number of negatives.

### B.5 EXPERIMENTAL SETTING FOR THE PPI DATASET

In the protein-protein interaction networks (PPI) dataset (Zitnik & Leskovec, 2017), a node represents a protein, and an edge represents an interaction between two proteins. Each node has 121 binary gene ontology (GO) terms, which describe biological information, such as biological pathways, cellular components, and molecular functions. For consistency with our experimental setup, we use the first GO term as the node label for prediction. The second GO term is used to determine the edge type. Edges connecting two nodes with the same second GO term are classified as type A, while edges connecting two nodes with different second GO terms are classified as type B. We then utilize the remaining 119 GO terms as node features for proteins, and apply structural missing with different rates. We use $80\%$ nodes for training, $10\%$ nodes for validation, and $10\%$ nodes for testing. We randomly create 10 different binary masks of a feature matrix for each $r_m$, and evaluate imputation methods across 10 runs using the generated masks. We consistently utilize GCN (Kipf & Welling, 2016a) models for downstream networks across imputation methods.

---

[6]https://github.com/liangchundong/HGNN-AC

## B.6 HYPERPARAMETER DETAILS

Table 5: Hyper-parameters ($\alpha$ and $\beta$) of HetGFD used in experiments with $r_m = 0.995$.

| Task | GNN | Missing type | Structural missing | | | Uniform missing | | |
|---|---|---|---|---|---|---|---|---|
| | | Dataset | ACM | DBLP | IMDB | ACM | DBLP | IMDB |
| Semi-supervised node classification | HGT | $\alpha$ | 0.7 | 0.1 | 0.7 | 0.1 | 0.1 | 0.3 |
| | | $\beta$ | 0.2 | 0.4 | 0.1 | 0.4 | 0.4 | 0.5 |
| | HGNN-AC | $\alpha$ | 0.7 | 0.1 | 0.7 | 0.5 | 0.1 | 0.5 |
| | | $\beta$ | 0.8 | 0.8 | 0.5 | 0.9 | 0.9 | 0.99 |
| Link prediction | HGT | $\alpha$ | 0.1 | 0.5 | 0.9 | 0.1 | 0.9 | 0.9 |
| | | $\beta$ | 0.4 | 0.8 | 0.8 | 0.5 | 0.9 | 0.99 |

Table 6: Hyper-parameters ($\alpha$ and $\beta$) of HetGFD under structural-missing settings with HGNN-AC and $r_m \in \{0.9, 0.5\}$.

| $r_m$ | Dataset | ACM | DBLP | IMDB |
|---|---|---|---|---|
| 0.9 | $\alpha$ | 0.7 | 0.1 | 0.9 |
| | $\beta$ | 0.8 | 0.99 | 0.2 |
| 0.5 | $\alpha$ | 0.9 | 0.1 | 0.9 |
| | $\beta$ | 0.99 | 0.99 | 0.8 |

Table 7: Hyper-parameters ($\alpha$ and $\beta$) of HetGFD on the PPI dataset.

| $r_m$ | 0.5 | 0.9 | 0.995 |
|---|---|---|---|
| $\alpha$ | 0.5 | 0.5 | 0.7 |
| $\beta$ | 0.8 | 0.8 | 0.4 |

To find the optimal hyperparameters $\alpha$ and $\beta$ for HetGFD, we perform a grid search on validation sets. The search range is set to $\{(\alpha, \beta)|\alpha \in \{0.9, 0.7, 0.5, 0.3, 0.1\}, \beta \in \{0.99, 0.9, 0.8, 0.5, 0.4, 0.2, 0.1, 0.05\}\}$. We set the value of K to 100. We list the hyperparameters of HetGFD used for HetGFD in our paper in Table 5, Table 6, and Table 7.

## C HYPERPARAMETER SENSITIVITY

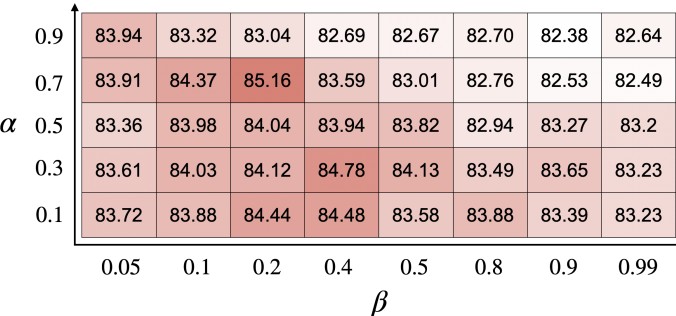

Figure 5: Average Macro-F1 score (%) in semi-supervised node classification for different $(\alpha, \beta)$ on ACM under a structural-missing setting with $r_m = 0.995$. HGT is commonly used as a downstream GNN. This heatmap shows the results on the validation set, where we search $\alpha$ and $\beta$.

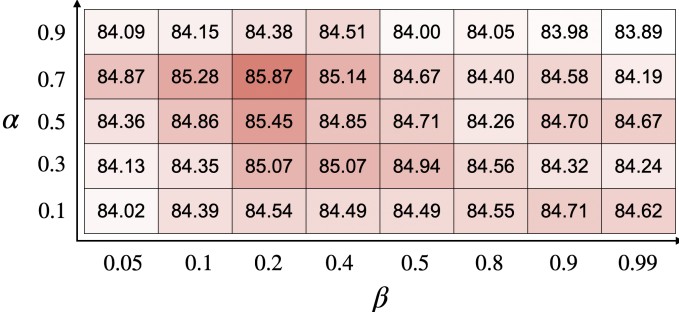

Figure 6: Average Macro-F1 score (%) in semi-supervised node classification for different $(\alpha, \beta)$ on ACM under a structural-missing setting with $r_m = 0.995$. HGT is commonly used as a downstream GNN. This heatmap shows the results on the test set.

| | 0.05 | 0.1 | 0.2 | 0.4 | 0.5 | 0.8 | 0.9 | 0.99 |
|---|---|---|---|---|---|---|---|---|
| 0.9 | 75.06 | 74.77 | 74.10 | 73.16 | 72.52 | 72.44 | 72.32 | 73.66 |
| 0.7 | 76.05 | 76.17 | 73.73 | 72.90 | 73.34 | 72.85 | 72.91 | 72.57 |
| 0.5 | 76.96 | 76.52 | 75.41 | 73.21 | 72.50 | 73.17 | 72.56 | 73.11 |
| 0.3 | 77.69 | 78.03 | 76.28 | 73.98 | 73.15 | 72.33 | 72.67 | 73.04 |
| 0.1 | 78.26 | 77.39 | 76.46 | 78.25 | 76.22 | 72.75 | 72.55 | 72.49 |

Figure 7: Average AUR ROC (%) in link prediction for different $(\alpha, \beta)$ on ACM under a structural-missing setting with $r_m = 0.995$. HGT is commonly used as a downstream GNN. This heatmap shows the results on the test set.

We conduct experiments to investigate the effects of hyperparameters of HetGFD. $\alpha$ and $\beta$ are hyperparameters of HetGFD, where $\alpha$ controls PC and $\beta$ determines the importance difference among edge types. Figure 5 shows the effects of the hyperparameters $\alpha$ and $\beta$ of HetGFD when HGT is used as a downstream GNN. We determine optimal hyperparameters based on the validation sets.

We conduct additional experiments on test sets in both semi-supervised node classification and link prediction. Figure 6 and Figure 7 show the results on ACM under structural-missing settings with a missing rate of $r_m = 99.5\%$. As shown in Figure 6, when the runner-up's Macro-F1 score is $84.47\%$, most combinations of $\alpha$ and $\beta$ achieve state-of-the-art performance. Similarly, as shown in Figure 7, when the runner-up's ROC AUC is $74.87\%$, many combinations achieve state-of-the-art performance, demonstrating the robustness of our HetGFD against $\alpha$ and $\beta$.

We conduct further experiments analyzing the impact of the diffusion step $K$ on performance. Table 8 shows the performance of HetGFD for different values of $K$, the number of diffusion steps. As shown in the table, while the performance of HetGFD with $K \in \{10, 20\}$ is slightly lower than others, the performance becomes stable after $K$ reaches 40. This is because imputed features approach the same steady state with a large enough $K$. In summary, $K \geq 40$ shows the robustness in both semi-supervised node classification and link prediction.

## D FURTHER EXPERIMENTS

### D.1 ABLATION STUDY

We conduct an extensive ablation study to analyze the effectiveness of the components in HetGFD. The experiments for semi-supervised node classification are performed under uniform-missing settings with $r_m = 0.995$ and HGT is commonly used.

Table 8: Performance of HetGFD for the different value of $K$, the number of diffusion steps.

**Semi-supervised node classification (Macro-F1)**

| $K$ | 10 | 20 | 40 | 100 (used) | 200 |
|---|---|---|---|---|---|
| ACM | $85.05 \pm 1.36$ | $85.92 \pm 0.81$ | $85.90 \pm 0.79$ | $85.87 \pm 0.81$ | $85.89 \pm 1.07$ |
| DBLP | $90.68 \pm 0.93$ | $90.84 \pm 0.81$ | $90.88 \pm 0.80$ | $90.88 \pm 0.79$ | $90.88 \pm 0.79$ |
| IMDB | $45.27 \pm 4.00$ | $46.33 \pm 2.96$ | $47.10 \pm 3.14$ | $47.15 \pm 1.66$ | $47.15 \pm 1.65$ |

**Link prediction (ROC AUC)**

| $K$ | 10 | 20 | 40 | 100 (used) | 200 |
|---|---|---|---|---|---|
| ACM | $77.90 \pm 1.12$ | $77.98 \pm 0.79$ | $78.10 \pm 0.88$ | $78.25 \pm 1.34$ | $78.25 \pm 1.34$ |
| DBLP | $91.80 \pm 0.54$ | $91.88 \pm 0.44$ | $91.98 \pm 0.50$ | $91.94 \pm 0.67$ | $91.97 \pm 0.52$ |
| IMDB | $91.45 \pm 1.15$ | $91.66 \pm 1.27$ | $92.50 \pm 1.04$ | $92.50 \pm 1.04$ | $92.50 \pm 1.04$ |

Table 9: Ablation study of HetGFD under uniform-missing settings with $r_m = 0.995$. $\triangle$ denotes using random edge-type ranking.

| PC | Edge-type ranking | ACM | | DBLP | | IMDB | |
|---|---|---|---|---|---|---|---|
| | | Macro-F1 | Micro-F1 | Macro-F1 | Micro-F1 | Macro-F1 | Micro-F1 |
| ✗ | ✗ | $84.07 \pm 1.38$ | $84.66 \pm 1.38$ | $89.47 \pm 1.76$ | $90.26 \pm 1.76$ | $45.57 \pm 2.03$ | $48.26 \pm 2.03$ |
| ✗ | ✓ | $85.19 \pm 1.04$ | $85.45 \pm 1.04$ | $89.89 \pm 1.62$ | $90.46 \pm 1.62$ | $46.25 \pm 2.27$ | $48.66 \pm 1.57$ |
| ✓ | ✗ | $86.26 \pm 0.69$ | $86.61 \pm 0.63$ | $90.49 \pm 1.43$ | $91.14 \pm 1.35$ | $47.25 \pm 1.88$ | $49.94 \pm 1.38$ |
| ✓ | △ | $86.87 \pm 2.15$ | $87.12 \pm 1.83$ | $90.20 \pm 1.50$ | $90.89 \pm 1.32$ | $47.13 \pm 2.94$ | $50.08 \pm 1.81$ |
| ✓ | ✓ | $\mathbf{88.14 \pm 0.69}$ | $\mathbf{88.14 \pm 0.66}$ | $\mathbf{90.65 \pm 1.53}$ | $\mathbf{91.40 \pm 1.24}$ | $\mathbf{48.57 \pm 1.41}$ | $\mathbf{50.57 \pm 1.66}$ |

Table 10: Performance comparison of HetGFD and its variant using direct edge-type-wise homophily ($\mathcal{H}$) instead of the edge ranking based on $\mathcal{H}$, in terms of Macro-F1 score on semi-supervised node classification.

| | ACM | DBLP | IMDB |
|---|---|---|---|
| direct $\mathcal{H}$ | $83.55 \pm 1.40$ | $89.25 \pm 1.65$ | $42.93 \pm 5.95$ |
| HetGFD | $\mathbf{85.87 \pm 0.81}$ | $\mathbf{90.88 \pm 0.79}$ | $\mathbf{47.15 \pm 1.66}$ |

**PC and edge-type ranking.** PC and edge-type ranking can be removed from HetGFD by substituting 1 for $\alpha$ and $\beta$, respectively. This substitution reduces HetGFD to FP, as it eliminates the mechanisms for controlling PC and distinguishing edge types. Additionally, HetGFD with random edge-type ranking is compared. The results of the ablation study are presented in Table 9. The results show that the performance gain achieved by edge-type ranking is significantly greater when PC is used than when it is not. This suggests that both PC and edge-type ranking mechanisms, which are grounded in the principle of feature homophily, work synergistically to boost performance. The notable performance improvement over the variant of HetGFD with random edge-type ranking further validates the concept of edge-type-wise homophily in determining edge-type ranking within our HetGFD framework.

**Direct $\mathcal{H}$ vs ranking-based.** We evaluate the performance of HetGFD using direct $\mathcal{H}$ values instead of the edge-type ranking approach. $\beta^{-(k-1)}$ in Eq. (6) and $\beta^{k-1}$ in Eq. (8) is where the edge-type ranking is utilized. $\beta^{-(k-1)}$ in Eq. (6) is designed to reduce the importance (PC) of features connected through low-ranking edges for the relation-aware diffusion stage, and $\beta^{k-1}$ in Eq. (8) is designed to make diffusion occur more through high-ranking edges for the relation-aware diffusion stage. We use the direct edge-type-wise homophily $\mathcal{H}$ of each edge type instead of $\beta^{-(k-1)}$ in Eq. (6) and $\beta^{k-1}$ in Eq. (8). We replace $\beta^{-(k-1)}$ in Eq. (6) with $\mathcal{H}^{-1}$ of each edge type and replace $\beta^{k-1}$ in Eq. (8) with $\mathcal{H}$ to align with the design concept of HetGFD. Table 10 shows the results. The results indicate that using direct $\mathcal{H}$ values results in lower performance across all datasets compared to when edge-type ranking is used. This suggests that transforming $\mathcal{H}$ into the ranking provides a more stable and effective way to leverage edge-type-wise homophily.

**Random initialization vs zero initialization.** We conduct additional experiments comparing the performance of HetGFD when zero initialization and random initialization are used for virtual fea-

Table 11: Performance of HetGFD for different initialization strategies for missing features. $K$ denotes the number of diffusion steps.

**Semi-supervised node classification (Macro-F1)**

| Initialization | ACM | DBLP | IMDB |
|---|---|---|---|
| random init ($K = 100$) | $85.81 \pm 0.78$ | $90.65 \pm 1.18$ | $46.47 \pm 2.06$ |
| random init ($K = 1000$) | $85.85 \pm 0.80$ | $90.90 \pm 0.80$ | $47.11 \pm 1.58$ |
| zero init (used) | $85.87 \pm 0.81$ | $90.88 \pm 0.79$ | $47.15 \pm 1.66$ |

**Link prediction (ROC AUC)**

| Initialization | ACM | DBLP | IMDB |
|---|---|---|---|
| random init ($K = 100$) | $77.57 \pm 1.91$ | $91.64 \pm 0.77$ | $92.50 \pm 1.04$ |
| random init ($K = 1000$) | $78.27 \pm 1.29$ | $91.96 \pm 0.74$ | $92.50 \pm 1.04$ |
| zero init (used) | $78.25 \pm 1.34$ | $91.94 \pm 0.67$ | $92.50 \pm 1.04$ |

ture generation. Table 11 shows the results. As shown in the table, zero initialization with $K = 100$ used in this paper shows slightly better performance compared to that of random initialization with $K = 100$, where $K$ is the number of diffusion steps. However, when we increase $K$ of HetGFD using random initialization to 1000, the performance of both becomes almost identical. This is because random initialization requires a larger value of $K$ to reach a steady state. Although updated features approach the same steady state with a large $K$ regardless of the initialization based on the proof in Appendix A, careful consideration is needed when determining $K$, depending on the initialization.

Table 12: Performance comparison of pre-imputed features obtained by preliminary diffusion and HetGFD's imputed matrix when fed to HGT.

**Semi-supervised node classification (Macro-F1)**

| | ACM | DBLP | IMDB |
|---|---|---|---|
| pre-imputed | $83.40 \pm 1.16$ | $88.92 \pm 1.82$ | $45.02 \pm 4.65$ |
| HetGFD | $\mathbf{85.87 \pm 0.81}$ | $\mathbf{90.88 \pm 0.79}$ | $\mathbf{47.15 \pm 1.66}$ |

**Link prediction (ROC AUC)**

| | ACM | DBLP | IMDB |
|---|---|---|---|
| pre-imputed | $72.53 \pm 2.10$ | $63.21 \pm 0.71$ | $63.47 \pm 13.05$ |
| HetGFD | $\mathbf{78.25 \pm 1.34}$ | $\mathbf{91.94 \pm 0.67}$ | $\mathbf{92.50 \pm 1.04}$ |

$\bar{\mathbf{X}}$ **vs** $\hat{\mathbf{X}}'$. To quantitatively assess the impact of our preliminary diffusion approach, we conduct additional experiments comparing the performance of semi-supervised node classification and link prediction tasks with pre-imputed features obtained via preliminary diffusion and HetGFD's imputed matrix. The results, presented in Table 12, show significant improvements when using HetGFD over pre-imputed features. Specifically, the Macro-F1 scores and ROC AUC metrics demonstrate that HetGFD consistently outperforms the preliminary diffusion approach across various datasets (ACM, DBLP, and IMDB). These results highlight the importance of edge-type-wise homophily and the resulting edge-type rankings.

Table 13: Performance comparison of HetGFD and its variant using uniform weights instead of $|\mathcal{E}_r|^{-1}$ in Eq. (2), in terms of Macro-F1 score on semi-supervised node classification. $\mathcal{E}_r$ denotes the set of $r$-type edges.

| | ACM | DBLP | IMDB |
|---|---|---|---|
| uniform weights | $\mathbf{85.87 \pm 0.81}$ | $90.29 \pm 1.68$ | $43.19 \pm 5.38$ |
| HetGFD | $\mathbf{85.87 \pm 0.81}$ | $\mathbf{90.88 \pm 0.79}$ | $\mathbf{47.15 \pm 1.66}$ |

**Uniform weights vs** $|\mathcal{E}_\mathbf{r}|^{-1}$ **in Eq. (2).** We compare the performance of HetGFD using uniform weights against the proposed $|\mathcal{E}_r|^{-1}$ weights in Eq. (2), where $\mathcal{E}_r$ denotes the set of $r$-type edges.

The results are shown in Table 13. While both variants perform equally well on ACM, HetGFD with the proposed weights achieved superior performance on DBLP and IMDB. This performance difference stems from that uniform weights cause biased diffusion toward the majority edge type. For example, in DBLP, there are edge types of Paper-Author, Paper-Term, and Paper-Conference, with each edge type having 39,290, 171,620, and 28,656 edges, respectively. In this case, using uniform weights causes excessive diffusion through Paper-Term edges, subsequently making the Paper-Term edge-type-wise homophily the highest. Since authors connected through Paper-Author edges may have more similar features than those connected through Paper-Term and Paper-Conference edges, using uniform weights leads to performance degradation. In contrast, we confirm that the original HetGFD consistently ranks Paper-Author as the highest edge-type-wise homophily.

Table 14: Performance comparison of HetGFD by assigning different weights to $\bar{\mathbf{W}}^{(d,r)}$ in Eq. (8), in terms of Macro-F1 score on semi-supervised node classification.

| $\bar{\mathbf{W}}^{(d,r)}$ in Eq. (8) | ACM | DBLP | IMDB |
|---|---|---|---|
| $\beta^{k-1}$ | $83.06 \pm 1.14$ | $88.57 \pm 2.83$ | $43.84 \pm 3.98$ |
| $\xi_{j,d}/\xi_{i,d}$ | $85.09 \pm 1.54$ | $90.29 \pm 1.12$ | $42.97 \pm 5.46$ |
| $\beta^{k-1} \cdot \xi_{j,d}/\xi_{i,d}$ (HetGFD) | $\mathbf{85.87 \pm 0.81}$ | $\mathbf{90.88 \pm 0.79}$ | $\mathbf{47.15 \pm 1.66}$ |

**Ablation study in Eq. (8).** We experiment with different weight assignments for $\bar{\mathbf{W}}^{(d,r)}$ in Eq. (8). In this equation, the term $\xi_{j,d}/\xi_{i,d}$ strengthens the message passing from high-PC features to low-PC features. The results in Table 3 show that the combination of $\beta^{k-1}$ and $\xi_{j,d}/\xi_{i,d}$ used in HetGFD consistently outperforms other weighting schemes. This highlights the effectiveness of our proposed weighting strategy.

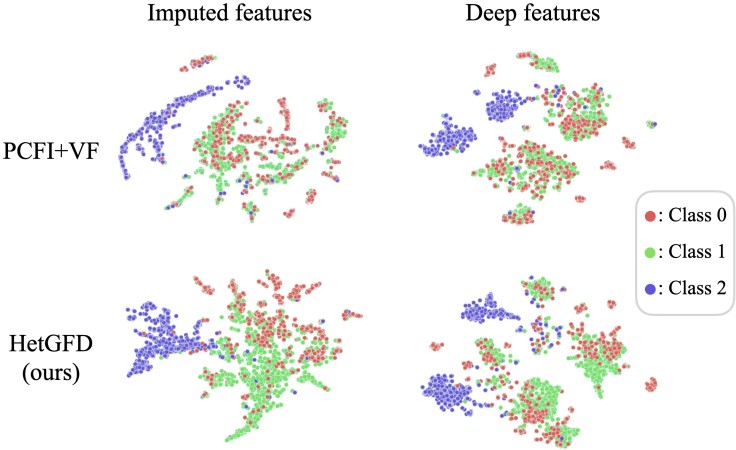

Figure 8: A t-SNE plot of imputed features and deep features learned with HGT.

## D.2 QUALITATIVE RESULTS

We provide qualitative results by employing t-SNE (Van der Maaten & Hinton, 2008) to visualize imputed features and deep features in HGT, obtained by HetGFD. For comparison, we also present visualizations of features obtained by PCFI+VF which is the most competitive method. Figure 8 shows the qualitative results on ACM under a structural-missing setting with $r_m = 0.995$. HetGFD provides much clearer cluster structures for both imputed features and deep features than PCFI+VF, indicating a more effective feature imputation.

## D.3 ADDITIONAL COMPLEXITY ANALYSIS

The two main processes to consider in the time complexity of HetGFD are the two diffusion stages, the calculation of edge-type-wise homophily $\mathcal{H}$ the calculation of PC. The two diffusion stage have a time complexity of $O(|\mathcal{E}|)$ since they are implemented using the message passing operation in

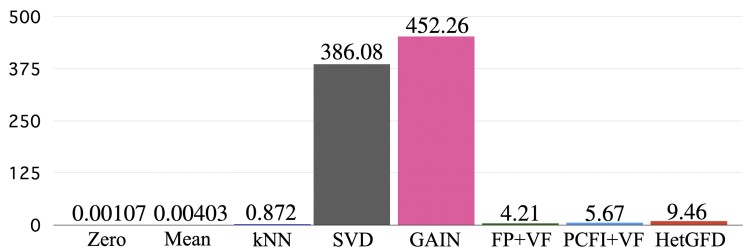

Figure 9: Imputation time (s) of methods on ACM under a structural-missing setting with $r_m = 0.995$.

PyTorch Geometric (Fey & Lenssen, 2019). This operation involves aggregating information from neighboring nodes, which scales linearly with the number of edges in the graph. The calculation of $\mathcal{H}$ has a time complexity of $O(F \cdot |\mathcal{E}|)$ since it measures feature similarity across all edges. The calculation of PC has a time complexity of $O(N^2)$. This is because the calculation of PC uses Dijkstra's algorithm, an algorithm for finding the shortest paths between nodes in a graph, for distance encoding. In structural-missing settings, the missing status of nodes is identical across all channels. Therefore, only a single transition matrix needs to be computed, requiring just one PC calculation. However, in uniform-missing settings, the missing status of nodes varies for each channel, necessitating different transition matrices for each of the $F$ channels. Consequently, $F$ PC calculations are required. Thus, HetGFD operates in structural-missing settings with a time complexity of $O(F \cdot |\mathcal{E}| + N^2)$ and in uniform-missing settings with a complexity of $O(F \cdot |\mathcal{E}| + F \cdot N^2)$. However, it is important to note that PCFI, the most competitive baseline, also involves the calculation of PC using Dijkstra's algorithm, which has a time complexity of $O(N^2)$. Compared to PCFI, our HetGFD consistently demonstrates superiority across all experimental settings.

For each imputation method, we measure the time for imputation. Figure 9 shows imputation time on a single split of ACM under a structural-missing setting with $r_m = 0.995$. While Zero and Mean show much shorter imputation time than the other methods, Zero and Mean suffer from significant performance degradation at high $r_m$. Meanwhile, diffusion-based methods (FP+VF, PCFI+VF, and HetGFD) show far less imputation time than SVD and GAIN. Although our method, HetGFD, necessitates more imputation time compared to other diffusion-based methods, it results in notable performance improvements in two major graph learning tasks.

To validate the scalability of HetGFD, we conduct additional experiments on OGBN-Arxiv (Hu et al., 2020a), which consists of 169343 nodes, 1166243 edges, and 128-dim node features. The nodes represent papers published in 2017, 2018, and 2019. We divide edges into three types: 2017-2018, 2018-2019, and 2017-2019. Under a structural-missing setting with $r_m$, we observed that HetGFD require only 0.39 hours for imputation. This demonstrates the efficiency of HetGFD in terms of imputation time, even for graphs with a large number of nodes.

### D.4 COMPARISON WITH FEATURE DISTANCE-BASED WEIGHTS

We conduct additional experiments to compare the weighted matrix $\bar{\mathbf{W}}^{(d)}$ used in relation-aware diffusion against feature distance-based weights. The purpose of $\bar{\mathbf{W}}^{(d)}$ is to assign greater weights to edge types that significantly contribute to feature homophily, *i.e.*, edge types that result in highly similar features between connected nodes. In contrast, the feature distance-based approach can be a simpler method for calculating weights. To construct the feature distance-based propagation matrix, we follow these steps: First, a pre-imputed feature matrix is generated using preliminary diffusion, as in HetGFD. Then, a single feature mean vector is computed for each node type based on the pre-imputed matrix. For each edge type, the feature distance is calculated by measuring the distance between the mean feature vectors of the two node types connected by that edge type. For each edge type, the calculated distance is inverted and assigned to the edges belonging to that edge type. Finally, the resulting weighted matrix is row-stochastically normalized to perform diffusion-based imputation.

Table 15 presents a performance comparison between the feature distance-based approach and our HetGFD. For this comparison, we perform semi-supervised classification and link prediction tasks,

Table 15: Performance comparison with HetGFD using feature distance-based weights.

**Semi-supervised node classification (Structural missing)**

| Missing Type | ACM | | DBLP | | IMDB | |
|---|---|---|---|---|---|---|
| | Macro-F1 | Micro-F1 | Macro-F1 | Micro-F1 | Macro-F1 | Micro-F1 |
| Distance-based | $83.30 \pm 1.45$ | $83.94 \pm 1.52$ | $88.48 \pm 2.30$ | $89.37 \pm 2.07$ | $43.23 \pm 3.83$ | $47.27 \pm 1.96$ |
| HetGFD | $\mathbf{85.87 \pm 0.81}$ | $\mathbf{86.10 \pm 0.81}$ | $\mathbf{90.88 \pm 0.79}$ | $\mathbf{91.53 \pm 0.79}$ | $\mathbf{47.15 \pm 1.66}$ | $\mathbf{49.46 \pm 1.66}$ |

**Semi-supervised node classification (Uniform missing)**

| Missing Type | ACM | | DBLP | | IMDB | |
|---|---|---|---|---|---|---|
| | Macro-F1 | Micro-F1 | Macro-F1 | Micro-F1 | Macro-F1 | Micro-F1 |
| Distance-based | $83.97 \pm 1.63$ | $84.70 \pm 1.35$ | $85.28 \pm 7.31$ | $87.28 \pm 3.95$ | $43.50 \pm 2.90$ | $47.39 \pm 1.62$ |
| HetGFD | $\mathbf{88.14 \pm 0.69}$ | $\mathbf{88.14 \pm 0.66}$ | $\mathbf{90.65 \pm 1.53}$ | $\mathbf{91.40 \pm 1.24}$ | $\mathbf{48.57 \pm 1.41}$ | $\mathbf{50.57 \pm 1.66}$ |

**Link prediction (Structural missing)**

| Missing Type | ACM | | DBLP | | IMDB | |
|---|---|---|---|---|---|---|
| | AUC | AP | AUC | AP | AUC | AP |
| Distance-based | $72.52 \pm 1.03$ | $73.21 \pm 1.37$ | $60.52 \pm 10.23$ | $59.43 \pm 9.59$ | $72.00 \pm 16.41$ | $67.66 \pm 12.23$ |
| HetGFD | $\mathbf{78.25 \pm 1.34}$ | $\mathbf{78.62 \pm 2.12}$ | $\mathbf{91.94 \pm 0.67}$ | $\mathbf{91.88 \pm 0.91}$ | $\mathbf{92.50 \pm 1.04}$ | $\mathbf{86.99 \pm 1.58}$ |

**Link prediction (Uniform missing)**

| Missing Type | ACM | | DBLP | | IMDB | |
|---|---|---|---|---|---|---|
| | AUC | AP | AUC | AP | AUC | AP |
| Distance-based | $72.37 \pm 1.27$ | $73.03 \pm 1.68$ | $60.40 \pm 9.66$ | $59.48 \pm 8.92$ | $70.57 \pm 18.07$ | $67.48 \pm 15.57$ |
| HetGFD | $\mathbf{76.96 \pm 1.74}$ | $\mathbf{77.19 \pm 1.98}$ | $\mathbf{92.17 \pm 0.56}$ | $\mathbf{92.12 \pm 0.53}$ | $\mathbf{91.95 \pm 1.72}$ | $\mathbf{86.72 \pm 3.40}$ |

consistently utilizing HGT models as downstream GNNs. As shown in the table, HetGFD significantly outperforms the feature distance-based approach across all settings. These results validate the effectiveness of designing the propagation matrix based on edge-type-wise homophily $\mathcal{H}$.

## D.5 STATISTICAL ANALYSIS

Table 16: $p$-values comparing our HetGFD to the runner-up in each setting. * denotes state-of-the-art, not a runner-up.

**For Table 1**

| GNN | Missing Type | Dataset | Metric | Runner-up | Runner-up's | Ours | $p$-value |
|---|---|---|---|---|---|---|---|
| HGT | Structural | ACM | Macro-F1 | PCFI+VF | $84.47 \pm 1.22$ | $\mathbf{85.87 \pm 0.81}$ | $5.98 \times 10^{-3}$ |
| | | | Micro-F1 | PCFI+VF | $85.06 \pm 0.98$ | $\mathbf{86.10 \pm 0.81}$ | $7.29 \times 10^{-3}$ |
| | | DBLP | Macro-F1 | PCFI+VF | $90.41 \pm 1.23$ | $\mathbf{90.88 \pm 0.79}$ | $2.23 \times 10^{-1}$ |
| | | | Micro-F1 | PCFI+VF | $91.14 \pm 1.08$ | $\mathbf{91.53 \pm 0.79}$ | $2.37 \times 10^{-1}$ |
| | | IMDB | Macro-F1 | PCFI+VF | $46.64 \pm 1.51$ | $\mathbf{47.15 \pm 1.66}$ | $5.30 \times 10^{-1}$ |
| | | | Micro-F1 | PCFI+VF | $\mathbf{49.69 \pm 1.83^{*}}$ | $49.46 \pm 1.66$ | $7.58 \times 10^{-1}$ |
| | Uniform | ACM | Macro-F1 | PCFI+VF | $86.26 \pm 0.69$ | $\mathbf{88.14 \pm 0.69}$ | $1.11 \times 10^{-3}$ |
| | | | Micro-F1 | PCFI+VF | $86.61 \pm 0.63$ | $\mathbf{88.14 \pm 0.66}$ | $1.84 \times 10^{-3}$ |
| | | DBLP | Macro-F1 | Mean | $90.54 \pm 1.62$ | $\mathbf{90.65 \pm 1.53}$ | $8.63 \times 10^{-1}$ |
| | | | Micro-F1 | Mean | $91.32 \pm 1.62$ | $\mathbf{91.40 \pm 1.24}$ | $9.27 \times 10^{-1}$ |
| | | IMDB | Macro-F1 | PCFI+VF | $47.25 \pm 1.88$ | $\mathbf{48.57 \pm 1.41}$ | $4.42 \times 10^{-2}$ |
| | | | Micro-F1 | PCFI+VF | $49.94 \pm 1.38$ | $\mathbf{50.57 \pm 1.66}$ | $2.63 \times 10^{-1}$ |

**For Table 18**

| GNN | Missing Type | Dataset | Metric | Runner-up | Runner-up's | Ours | $p$-value |
|---|---|---|---|---|---|---|---|
| HGNN-AC | Structural | ACM | Macro-F1 | PCFI+VF | $69.25 \pm 4.32$ | $\mathbf{76.23 \pm 2.84}$ | $1.56 \times 10^{-4}$ |
| | | | Micro-F1 | PCFI+VF | $75.07 \pm 4.32$ | $\mathbf{76.91 \pm 2.84}$ | $1.21 \times 10^{-1}$ |
| | | DBLP | Macro-F1 | PCFI+VF | $93.02 \pm 0.49$ | $\mathbf{93.26 \pm 0.40}$ | $2.49 \times 10^{-1}$ |
| | | | Micro-F1 | PCFI+VF | $93.49 \pm 0.49$ | $\mathbf{93.76 \pm 0.40}$ | $1.59 \times 10^{-1}$ |
| | | IMDB | Macro-F1 | PCFI+VF | $34.29 \pm 3.46$ | $\mathbf{35.05 \pm 1.84}$ | $5.98 \times 10^{-1}$ |
| | | | Micro-F1 | PCFI+VF | $42.99 \pm 3.46$ | $\mathbf{43.64 \pm 1.84}$ | $5.62 \times 10^{-1}$ |
| | Uniform | ACM | Macro-F1 | PCFI+VF | $84.04 \pm 2.08$ | $\mathbf{85.27 \pm 1.61}$ | $7.72 \times 10^{-2}$ |
| | | | Micro-F1 | PCFI+VF | $84.92 \pm 3.26$ | $\mathbf{85.64 \pm 1.61}$ | $4.79 \times 10^{-1}$ |
| | | DBLP | Macro-F1 | PCFI+VF | $93.77 \pm 0.44$ | $\mathbf{94.03 \pm 0.29}$ | $1.63 \times 10^{-1}$ |
| | | | Micro-F1 | PCFI+VF | $94.21 \pm 0.44$ | $\mathbf{94.45 \pm 0.29}$ | $1.26 \times 10^{-1}$ |
| | | IMDB | Macro-F1 | PCFI+VF | $41.06 \pm 3.87$ | $\mathbf{43.52 \pm 2.66}$ | $8.30 \times 10^{-2}$ |
| | | | Micro-F1 | PCFI+VF | $47.20 \pm 3.87$ | $\mathbf{47.87 \pm 2.66}$ | $4.30 \times 10^{-1}$ |

**For Table 2**

| Missing Type | Dataset | Metric | Runner-up | Runner-up's | Ours | $p$-value |
|---|---|---|---|---|---|---|
| Structural | ACM | AUC | GRAFENNE | $74.87 \pm 6.71$ | $\mathbf{78.25 \pm 1.34}$ | $1.58 \times 10^{-1}$ |
| | | AP | FP+VF | $74.03 \pm 0.84$ | $\mathbf{78.62 \pm 2.12}$ | $2.65 \times 10^{-4}$ |
| | DBLP | AUC | GRAFENNE | $90.14 \pm 7.26$ | $\mathbf{91.94 \pm 0.67}$ | $4.50 \times 10^{-1}$ |
| | | AP | GRAFENNE | $76.53 \pm 7.12$ | $\mathbf{91.88 \pm 0.91}$ | $1.08 \times 10^{-4}$ |
| | IMDB | AUC | FP+VF | $\mathbf{92.50 \pm 1.04}$ | $\mathbf{92.50 \pm 1.04}$ | $1$ |
| | | AP | FP+VF | $\mathbf{86.99 \pm 1.58}$ | $\mathbf{86.99 \pm 1.58}$ | $1$ |
| Uniform | ACM | AUC | PCFI+VF | $74.94 \pm 1.37$ | $\mathbf{76.96 \pm 1.74}$ | $1.60 \times 10^{-2}$ |
| | | AP | PCFI+VF | $73.80 \pm 1.63$ | $\mathbf{77.19 \pm 1.98}$ | $7.98 \times 10^{-3}$ |
| | DBLP | AUC | GAIN | $72.48 \pm 0.62$ | $\mathbf{92.17 \pm 0.56}$ | $4.57 \times 10^{-15}$ |
| | | AP | GAIN | $74.20 \pm 0.60$ | $\mathbf{92.12 \pm 0.53}$ | $5.39 \times 10^{-13}$ |
| | IMDB | AUC | GAIN | $\mathbf{92.50 \pm 1.04^{*}}$ | $91.95 \pm 1.72$ | $4.98 \times 10^{-1}$ |
| | | AP | GAIN | $\mathbf{86.99 \pm 1.58^{*}}$ | $86.72 \pm 3.40$ | $8.47 \times 10^{-1}$ |

**For Table 3**

| $r_m$ | Runner-up | Runner-up's | Ours | $p$-value |
|---|---|---|---|---|
| 0.5 | PCFI+VF | $80.75 \pm 1.68$ | $\mathbf{81.57 \pm 1.04}$ | $2.53 \times 10^{-1}$ |
| 0.9 | PCFI+VF | $65.22 \pm 2.19$ | $\mathbf{66.84 \pm 1.92}$ | $7.68 \times 10^{-2}$ |
| 0.995 | GRAFENNE | $62.26 \pm 0.00$ | $\mathbf{63.20 \pm 0.37}$ | $3.25 \times 10^{-5}$ |

We conduct additional experiments to evaluate the statistical significance of our HetGFD's superior performance. Table 16 shows $p$-values comparing EDBD to the runner-up in each setting for all the results in Table 1, Table 18, Table 2, and Table 3. As shown in the table, the $p$-values for Table 1, Table 18, Table 2, and Table 3 exhibit a wide range of distributions. Nevertheless, while the runner-ups vary depending on the setting, HetGFD consistently achieves state-of-the-art performance, except for link prediction on DBLP under the uniform-missing setting.

Table 17: Normalized Dirichlet energy for each metapath.

| Dataset | Edge Type | Normalized Dirichlet energy |
|---------|-----------|------------------------------|
| ACM | Paper-Author-Paper | 0.105 |
|  | Paper-Subject-Paper | 0.149 |
| DBLP | Paper-Author-Paper | 0.159 |
|  | Paper-Term-Paper | 0.183 |
|  | Paper-Conference-Paper | 0.203 |
| IMDB | Movie-Director-Movie | 0.125 |
|  | Movie-Actor-Movie | 0.203 |

## E    DISCUSSION ON FEATURE HOMOPHILY

In the context of homophily, two types are commonly discussed: class homophily and feature homophily. Diffusion-based imputation methods rely on feature homophily. However, in heterogeneous graphs, some node types often lack features, making it very challenging to assess the overall level of feature homophily in such graphs (*i.e.*, whether the graph is feature-homophilic or non-feature-homophilic (feature-heterophilic)). Nevertheless, if one wishes to measure feature homophily, it can be done using metapaths (*e.g.*, Movie-Director-Movie or Movie-Actor-Movie) that connect node types with features. Each metapath exhibits a distinct level of feature homophily within a heterogeneous graph. For example, some metapaths, such as Movie-Director-Movie, exhibit high feature homophily, while others, like Movie-Actor-Movie, display low feature homophily.

Table 17 presents the normalized Dirichlet energy (lower values indicate higher feature homophily), representing the feature homophily levels for each metapath in the datasets used in this paper. As shown in the table, all datasets contain metapaths with varying levels of feature homophily. We emphasize that, while it is very challenging to determine whether a heterogeneous graph is feature-homophilic or feature-heterophilic, diverse levels of feature homophily exist within the graph depending on the metapaths. Through the extensive experiments, we confirm that our HetGFD effectively prevents performance degradation on downstream tasks under various missing data scenarios across these datasets as well as the PPI dataset. However, the effectiveness of HetGFD may be limited on feature-heterophilic heterogeneous graphs, where most paths connecting attributed nodes are heterophilic connections.

# F   ADDITIONAL EXPERIMENTAL RESULTS

Table 18: Semi-supervised node classification results (%) with $r_m = 0.995$.

| GNN | Missing type | Method | ACM | | DBLP | | IMDB | |
|---|---|---|---|---|---|---|---|---|
| | | | Macro-F1 | Micro-F1 | Macro-F1 | Micro-F1 | Macro-F1 | Micro-F1 |
| | | Full features | $89.09 \pm 2.38$ | $89.22 \pm 1.95$ | $93.48 \pm 0.58$ | $93.95 \pm 0.58$ | $55.23 \pm 3.07$ | $56.67 \pm 1.70$ |
| HGNN-AC | Structural missing | Zero | $42.14 \pm 6.12$ | $58.01 \pm 6.12$ | $13.14 \pm 4.97$ | $28.81 \pm 4.97$ | $19.34 \pm 0.44$ | $36.76 \pm 0.44$ |
| | | Mean | $21.86 \pm 0.04$ | $48.75 \pm 0.04$ | $11.33 \pm 0.98$ | $28.35 \pm 0.98$ | $19.61 \pm 0.78$ | $36.38 \pm 0.78$ |
| | | kNN | $42.45 \pm 6.33$ | $58.63 \pm 6.39$ | $13.40 \pm 4.88$ | $28.95 \pm 4.88$ | $19.34 \pm 0.44$ | $36.76 \pm 0.44$ |
| | | SVD | $41.59 \pm 4.63$ | $57.09 \pm 4.63$ | $12.04 \pm 1.93$ | $27.78 \pm 1.78$ | $19.34 \pm 0.44$ | $36.76 \pm 0.44$ |
| | | GAIN | $40.15 \pm 4.40$ | $56.23 \pm 4.40$ | $11.28 \pm 0.69$ | $27.78 \pm 2.17$ | $19.34 \pm 0.44$ | $36.76 \pm 0.44$ |
| | | GRAFENNE | $52.77 \pm 7.28$ | $65.49 \pm 5.17$ | $18.65 \pm 3.13$ | $27.73 \pm 0.94$ | $27.20 \pm 2.73$ | $36.12 \pm 1.76$ |
| | | FP+VF | $64.22 \pm 5.33$ | $73.00 \pm 5.33$ | $17.60 \pm 16.03$ | $32.96 \pm 16.03$ | $20.31 \pm 1.20$ | $37.01 \pm 1.20$ |
| | | PCFI+VF | $69.25 \pm 4.32$ | $75.07 \pm 4.32$ | $93.02 \pm 0.49$ | $93.49 \pm 0.49$ | $34.29 \pm 3.46$ | $42.99 \pm 3.46$ |
| | | HetGFD (ours) | $\mathbf{76.23 \pm 2.84}$ | $\mathbf{76.91 \pm 2.84}$ | $\mathbf{93.26 \pm 0.40}$ | $\mathbf{93.76 \pm 0.40}$ | $\mathbf{35.05 \pm 1.84}$ | $\mathbf{43.64 \pm 1.84}$ |
| | Uniform missing | Zero | $21.85 \pm 0.00$ | $48.74 \pm 0.00$ | $11.51 \pm 0.57$ | $27.76 \pm 1.42$ | $19.63 \pm 0.71$ | $36.84 \pm 0.71$ |
| | | Mean | $21.86 \pm 0.04$ | $48.75 \pm 0.04$ | $11.28 \pm 1.78$ | $27.32 \pm 2.06$ | $20.93 \pm 1.58$ | $36.42 \pm 1.58$ |
| | | kNN | $21.85 \pm 0.00$ | $48.74 \pm 0.00$ | $11.40 \pm 0.63$ | $27.51 \pm 1.50$ | $20.55 \pm 1.24$ | $36.55 \pm 1.24$ |
| | | SVD | $60.63 \pm 4.62$ | $68.11 \pm 4.62$ | $11.72 \pm 0.79$ | $27.54 \pm 1.73$ | $19.86 \pm 0.78$ | $36.86 \pm 0.78$ |
| | | GAIN | $55.28 \pm 10.33$ | $65.40 \pm 10.33$ | $11.38 \pm 0.57$ | $28.78 \pm 1.73$ | $19.63 \pm 0.71$ | $36.84 \pm 0.71$ |
| | | GRAFENNE | $73.04 \pm 1.34$ | $74.29 \pm 1.06$ | $17.8 \pm 2.24$ | $28.61 \pm 0.91$ | $36.51 \pm 1.88$ | $39.82 \pm 1.60$ |
| | | FP+VF | $83.27 \pm 2.04$ | $84.04 \pm 2.04$ | $11.95 \pm 2.59$ | $28.21 \pm 2.59$ | $20.61 \pm 1.97$ | $37.24 \pm 1.97$ |
| | | PCFI+VF | $84.04 \pm 2.08$ | $84.92 \pm 3.26$ | $93.77 \pm 0.44$ | $94.21 \pm 0.44$ | $41.06 \pm 3.87$ | $47.20 \pm 3.87$ |
| | | HetGFD (ours) | $\mathbf{85.27 \pm 1.61}$ | $\mathbf{85.64 \pm 1.61}$ | $\mathbf{94.03 \pm 0.29}$ | $\mathbf{94.45 \pm 0.29}$ | $\mathbf{43.52 \pm 2.66}$ | $\mathbf{47.87 \pm 2.66}$ |

