# OpenReview forum: "Relation-Aware Diffusion for Heterogeneous Graphs with Partially Observed Features"
_ICLR.cc/2025/Conference — ICLR 2025 Poster_

### Official Review · Reviewer_P2Yo · 2024-10-24

**Soundness:** 3
**Presentation:** 3
**Contribution:** 3
**Rating:** 6
**Confidence:** 4

**Summary:**

The paper proposes a new diffusion-based imputation method for heterogeneous graphs to improve classification. The method is applied to each dimension. Given a dimension d, it starts by creating a transition Matrix \tilde{M}^(d), where the rows corresponding to the known attributes are set to 1 in the main diagonal (no transition) and the other rows correspond to the probability of going from one node to another. This probability is proportional to the number of edges existing per type of edge. Once the method converges it generates the pre-imputed matrix which is used to compute the Edge-type-wise homophily ranking.

The Edge-type-wise homophily ranking corresponds to ranking from 1 to r where a value of 1 corresponds to the edge type with the highest homophily. The highest homophily corresponds to the average cosine similarity among nodes of each type of edge standardized by the expected similarity ("calculated from randomly sampled node pairs"). This ranking is used to generate a weight adjacency matrix given by equation 6 (high homophily implies lower weights), which is later used to estimate the pseudo-confidence of a node.

The pseudo-confidence of a node is given by $\alpha \times$ the shortest path of the node with respect to a node with a known attribute. So, a high pseudo-confidence implies that the node is closer and its value can be trusted. The pseudo-confidence is again used to estimate a new transition matrix to generate the final values in a new iterative process.

In summary, the first transition matrix determines "the importance of nodes based on their similarity/homophily" and the second transition matrix generates the final attributes based on the imputed importance of the nodes. Then, the imputed features are used for prediction.

**Strengths:**

The abstract and the introduction define the problem correctly. Even though it seems that the problem will be feature imputation, the problem is clearly defined as the generation of imputed features to improve classification rather than generating unobserved features.

The method converges thanks to the absorbing state defined by the known nodes. The convergence is also demonstrated in the appendix, but it should be easier to connect it to Markov random models.

The main contribution of the paper seems to be the inclusion of heterogeneous matrices, which are considered through the summation of the different dimensions.

**Weaknesses:**

The clarity of the paper is very low. The main idea of the paper can be explained with simple words and the process simplified. For some reason, the authors prefer to use a mathematical explanation instead of a simple algorithm or explanation. Most equations are left for the readers to understand, instead of explaining their meaning and their use in the next steps of the paper. These reduce the readability of the paper. Moreover, the idea of the transition matrix is quite similar to Um et al 2023. So, the paper could be rewritten based on the Um idea, and explaining that the main contribution is the "summation" of the matrices, leaving some technical details to the appendix or the Um paper.

The quality of the paper can also be improved. Besides the complex readability, the authors should try to explain why this method actually works. The paper should state the main difference in comparison to the PCFI method (UM paper). Moreover, the main paper lacks several components, including time complexity, experiment details, and hyperparameter analysis. Even though these parts are included in the Appendix, we are not required to read it for the evaluation of the paper.

Regarding the time complexity, the method is impossible to apply to a large network. As expected, the time complexity is proportional to |V|^2.

Based on the final results, it seems that the differences are not statistically significant. This reduces the significance of this method in comparison to PCFI. Even though the hyperparameters analysis is given in the appendix, the main paper does not discuss their values or their influence on the model. Moreover, no training process is given for these values (\alpha and \beta).

In summary, the paper is complex to read, the most important discussion and experiment details are left in the appendix, and it has a large time complexity.

**Questions:**

Could you give more details about the time complexity analysis?
Is the size of the network from Table 4 the main connected component?
What is the behavior when you include \alpha and \beta equal to 1?

---

> ### Author Response · Authors · 2024-11-24
> **Response to Reviewer P2Yo (1/3)**
>
> We sincerely thank the reviewer for their constructive feedback and insightful questions, which have significantly improved our work. We provide our responses below.
>
> >**W1.** The clarity of the paper is very low. The main idea of the paper can be explained with simple words and the process simplified. For some reason, the authors prefer to use a mathematical explanation instead of a simple algorithm or explanation. Most equations are left for the readers to understand, instead of explaining their meaning and their use in the next steps of the paper. These reduce the readability of the paper.
>
> **A1.** We acknowledge that the original manuscript relied heavily on equations without sufficient accompanying explanations. We appreciate the reviewer’s deep understanding despite the lack of detailed explanations in the original submission. **As the reviewer suggested, we have thoroughly restructured the proposed method section** in the revised manuscript, employing simpler and clearer language. **We have provided detailed step-by-step explanations for each equation**, including its meaning and usage. These revisions have significantly improved the readability of the paper. **We are confident that if the reviewer revisits the proposed method section, they will find it significantly much easier to understand, as will other readers**.
>
> >**W2.** Moreover, the idea of the transition matrix is quite similar to Um et al 2023. So, the paper could be rewritten based on the Um idea, and explaining that the main contribution is the "summation" of the matrices, leaving some technical details to the appendix or the Um paper.\The quality of the paper can also be improved. Besides the complex readability, the authors should try to explain why this method actually works. The paper should state the main difference in comparison to the PCFI method (UM paper).
>
> **A2.** The idea of the transition matrix may appear similar since we incorporate the concept of pseudo-confidence introduced by [1]. However, **there are two significant differences** between the transition matrix of [1] and that of our HetGFD.
>
> First, as explained in lines 298-302, **PC of [1] is calculated using an unweighted graph, whereas our HetGFD calculates PC on a weighted graph based on edge-type-wise homophily $\mathcal{H}$**. This design enables HetGFD to perform relation-aware distance encoding by leveraging the information inherent in the edge types of heterogeneous graphs. Second, as stated in lines 330-342, **we assign greater weights to edges belonging to a high-$\mathcal{H}$ edge type**. These two unique design aspects of HetGFD, which make HetGFD specifically tailored for heterogeneous graphs.
>
> **Our main contribution is not the summation of matrices**; both FP [2] + virtual features (VF) and PCFI [1] + VF, which are used as baselines, also sum edge-type-wise adjacency matrices across all edge types. However, they treat all edges equally, regardless of their types. **Our main contributions are (1) virtual features addressing non-attributed nodes and (2) two unique design aspects enabling relation-aware diffusion in heterogeneous graphs**. To the best of our knowledge, **HetGFD is the first diffusion-based imputation scheme tailored to heterogeneous graphs** while existing diffusion-based imputation methods cannot be directly applied to heterogeneous graphs.
>
> We initially structured the paper by pointing out the limitations of existing diffusion-based imputation methods [1, 2] and focusing on the differences between our approach and those methods. However, in response to the reviewer’s feedback, we have further revised lines 302-311 and 327-335 in the revised manuscript to make these differences more explicit. Additionally, we have included an explanation in lines 364-371 to clarify why HetGFD is effective.
>
> [1] Um, Daeho, et al. "Confidence-Based Feature Imputation for Graphs with Partially Known Features." ICLR 2023.\
> [2] Rossi, Emanuele, et al. "On the unreasonable effectiveness of feature propagation in learning on graphs with missing node features." Learning on graphs conference. PMLR, 2022.

---

> ### Author Response · Authors · 2024-11-24
> **Response to Reviewer P2Yo (2/3)**
>
> >**W3.** Moreover, the main paper lacks several components, including time complexity, experiment details, and hyperparameter analysis. Even though these parts are included in the Appendix, we are not required to read it for the evaluation of the paper.
>
> **A3.** While we strongly agree with the reviewer that time complexity, experiment details, and hyperparameter analysis are important, **the page limit for the main text at ICLR is 10 pages**, which often leads **many ICLR papers [1,2,3] to include components such as time complexity, experiment details, and hyperparameter analysis in the appendix** rather than the main text. We have attempted to streamline the main text as much as possible, it was challenging to secure space in the main text as it already contains only the most essential parts. **In response, we have summarized the time complexity analysis in Appendix D.3 and incorporated it into Sec. 6 of the main manuscript**. If the reviewer identifies a component in the revised manuscript's appendix that should be included in the main text, please let us know, and we will promptly incorporate it.
>
> [1] Zhou, Zhanke, et al. "Less is More: One-shot Subgraph Reasoning on Large-scale Knowledge Graphs." ICLR 2024.\
> [2] Zhou, Xiong, et al. "Variance-enlarged Poisson Learning for Graph-based Semi-Supervised Learning with Extremely Sparse Labeled Data." ICLR 2024.\
> [3] De Felice, Giovanni, et al. "Graph-based Virtual Sensing from Sparse and Partial Multivariate Observations." ICLR 2024.
>
> >**W4.** Regarding the time complexity, the method is impossible to apply to a large network. As expected, the time complexity is proportional to |V|^2.\
> **Q1.** Could you give more details about the time complexity analysis?
>
> **A4.** To address the reviewer’s concern regarding HetGFD’s complexity, we have conducted an in-depth analysis by examining each process that comprises HetGFD.
>
> The main processes to consider in the time complexity of HetGFD are the two diffusion stages, the calculation of edge-type-wise homophily $\mathcal{H}$ the calculation of PC. The two diffusion stage have a time complexity of $O(|\mathcal{E}|)$ since they are implemented using the message passing operation in PyTorch Geometric [4]. This operation involves aggregating information from neighboring nodes, which scales linearly with the number of edges in the graph. The calculation of $\mathcal{H}$ has a time complexity of $O(F\cdot|\mathcal{E}|)$ since it measures feature similarity across all edges. The calculation of PC has a time complexity of $O(N^2)$. This is because the calculation of PC uses Dijkstra's algorithm, an algorithm for finding the shortest paths between nodes in a graph, for distance encoding. In structural-missing settings, the missing status of nodes is identical across all channels. Therefore, only a single transition matrix needs to be computed, requiring just one PC calculation. However, in uniform-missing settings, the missing status of nodes varies for each channel, necessitating different transition matrices for each of the $F$ channels. Consequently, $F$ PC calculations are required. Thus, HetGFD operates **in structural-missing settings with a time complexity of $O(F\cdot|\mathcal{E}|+N^2)$** and **in uniform-missing settings with a complexity of $O(F\cdot|\mathcal{E}|+F \cdot N^2)$**. However, it is important to note that **PCFI, the most competitive baseline, also involves the calculation of PC using Dijkstra's algorithm, which has a time complexity of $O(N^2)$**. Compared to PCFI, **our HetGFD consistently demonstrates superiority** across all experimental settings.
>
> We have included a detailed explanation of HetGFD's complexity in Appendix D.3 of the revised manuscript.
>
> [4] Fey, Matthias, and Jan Eric Lenssen. "Fast graph representation learning with PyTorch Geometric." arXiv preprint arXiv:1903.02428 (2019).

---

> ### Author Response · Authors · 2024-11-24
> **Response to Reviewer P2Yo (3/3)**
>
> >**W5.** Based on the final results, it seems that the differences are not statistically significant. This reduces the significance of this method in comparison to PCFI.
>
> **A5.** To address the reviewer’s concern, **we conduct additional experiments to evaluate the statistical significance** of our HetGFD's superior performance. Table 16 in the revised manuscript shows **$p$-values comparing EDBD to the runner-up in each setting for all the results in Table 1, Table 18, Table 2, and Table 3**. As shown in the table, the $p$-values for Table 1, Table 18, Table 2, and Table 3 exhibit a wide range of distributions. Nevertheless, **while the runner-ups vary depending on the setting, HetGFD consistently achieves state-of-the-art performance**, except for link prediction on DBLP under the uniform-missing setting. We have added this discussion regarding statistical significance to Appendix D.5 in the revised manuscript.
>
> **Table 16**: https://anonymous.4open.science/r/ICLR_2025_6591_tables-E865/Table%2016.png
>
> >**Q2.** Is the size of the network from Table 4 the main connected component?
>
> **A6.** **Yes**, the dataset statistics in Table 4, including the network size, represent the main connected component where the experiments are conducted.
>
> >**Q3.** What is the behavior when you include \alpha and \beta equal to 1?
>
> **A7.** **Setting $\alpha$ and $\beta$ to 1 reduces HetGFD to FP**. This is because $\alpha$ in Definition 2 controls the adjustment of pseudo-confidence (PC), and $\beta$ governs edge-type-wise distinction in relation-aware diffusion. When both are set to 1, these mechanisms are disabled, resulting in standard FP behavior. In Appendix D.1, we set $\alpha$ and $\beta$ to 1 to remove the effects of PC and edge-type ranking for the ablation study. **We have clarified that this behavior reduces HetGFD to FP in lines 1059-1060 of the revised manuscript**.

---

> > ### Comment · Reviewer_P2Yo · 2024-11-25
> >
> > Thanks for your response, you considered all the comments, and the new version of the paper seems better to me than before. The \alpha and \beta question was with that purpose; so people can understand the relation with the other models too. I increased my score, thanks!

---

> > > ### Author Response · Authors · 2024-11-27
> > > **Request for Clarification Regarding Presentation Score**
> > >
> > > Dear Reviewer P2Yo,
> > >
> > > We sincerely appreciate your initial feedback regarding readability and your recognition of the improvements reflected in your increased rating. Your valuable feedback has significantly enhanced the quality of our paper, and we are truly grateful for your thoughtful input.
> > >
> > > We noticed that the presentation score has remained the same, and we were wondering if there is a chance it might not have been updated by oversight. If the presentation score was intentionally kept as is when you updated your rating, we completely respect your decision.
> > >
> > > Thank you once again for your time and constructive feedback.
> > >
> > > Sincerely,\
> > > The Authors

---

> > > > ### Author Response · Authors · 2024-12-02
> > > > **[Gentle Reminder] Kindly Seeking Your Response Before the Deadline**
> > > >
> > > > Dear **Reviewer P2Yo**,
> > > >
> > > > With only **21 hours remaining before the deadline**, we kindly ask you to take a moment to review our previous inquiry regarding the presentation score. As mentioned earlier, we sincerely appreciate your professional feedback and your thoughtful reevaluation of our work. Your acknowledgment of our rebuttal, which focuses on enhancing readability, and the increased rating suggest that the presentation score might not have been updated due to an oversight. If the presentation score was intentionally left unchanged, we completely respect your decision.
> > > >
> > > > Thank you once again for your time and thoughtful review.
> > > >
> > > > Sincerely,\
> > > > The Authors

---

> > > > > ### Comment · Reviewer_P2Yo · 2024-12-02
> > > > >
> > > > > I already changed my score and increased it days ago.

---

> > > > > > ### Author Response · Authors · 2024-12-02
> > > > > >
> > > > > > Dear **Reviewer P2Yo**,
> > > > > >
> > > > > > Thank you for your prompt response. We have confirmed that your overall rating has been increased to 6, and we sincerely appreciate your updated evaluation. However, our inquiry specifically concerns the detailed score for Presentation, which is currently rated as 1: poor. If the presentation score was intentionally left unchanged, we apologize for bringing up the request.
> > > > > >
> > > > > > Thank you once again for your time and consideration.
> > > > > >
> > > > > > Sincerely,\
> > > > > > The Authors

---

> > > > > > > ### Comment · Reviewer_P2Yo · 2024-12-02
> > > > > > >
> > > > > > > Ok, I changed.

---

> > > > > > > > ### Author Response · Authors · 2024-12-02
> > > > > > > > **Appreciation for Acknowledgment of Presentation Improvement**
> > > > > > > >
> > > > > > > > Dear **Reviewer P2Yo**,
> > > > > > > >
> > > > > > > > Thank you for **updating the presentation score to 3: good**. Your professional feedback has significantly improved the presentation, which was initially a weaker aspect of our paper, elevating it to the same high level as its soundness and contribution.
> > > > > > > >
> > > > > > > > We sincerely appreciate your support and consideration.
> > > > > > > >
> > > > > > > > Sincerely,\
> > > > > > > > The Authors

---

> ### Comment · Area_Chair_HUP6 · 2024-11-25
>
> Could please acknowledge and respond to the rebuttal.

---

> ### Author Response · Authors · 2024-11-26
>
> Thank you for taking the time to review our rebuttal and for increasing your score. We greatly appreciate your recognition of the improvements in the revised manuscript. Your feedback on readability has significantly enhanced the quality of our paper. Additionally, your insights, particularly regarding $\alpha$ and $\beta$, have been invaluable in helping us better emphasize the relationship with other models and clarify the distinctions between our model and theirs. If there are any remaining concerns, please let us know; we are fully prepared to address them immediately.

---

### Official Review · Reviewer_QsiT · 2024-11-02

**Soundness:** 2
**Presentation:** 3
**Contribution:** 3
**Rating:** 6
**Confidence:** 3

**Summary:**

This paper proposes a method for feature imputation in heterogeneous graphs. It is assumed that only some types of nodes may have a certain feature and for some of them the feature value is not given and has to be imputed. The idea is the following. First, virtual features are assigned to nodes for which a particular feature is undefined. Then, preliminary propagation is used to get initial assignments for all the nodes. Using these assignments, homophily of different edge types is defined. Then, these homophily values are used to re-weight the edges for the final propagation of features. In the experiments, it is shown that the proposed approach outperforms the baselines for different missing feature rates.

**Strengths:**

- The paper is in general well-written and easy to follow;
- The proposed approach, while being relatively simple, shows good results in the experiments;
- The authors conduct extensive ablation studies to demonstrate that different components of their solution are important.

**Weaknesses:**

- It seems that there are two assumptions needed for the approach to work: 1) the graph should be homophilic (nodes with small graph distances between them have similar features) so that feature propagation works, and 2) features are of the same nature/scale so that equation (5) is meaningful. These limitations should be discussed.
- Some of the components of the proposed solution are designed based on some intuition, but particular formulas are not well motivated (e.g., by models or theoretical analysis). Examples include a particular weighting of the edge types in eq. (2), ranking edge types according to homophily and eq. (6), and equation (8). I acknowledge the ablation studies in Appendix, but still many potential solutions could be used and it would be insightful to understand why a particular form is chosen.
- One of the baselines (PCFI+VF) shows comparable performance in most cases, I think it would be useful to discuss this method and compare its advantages and disadvantages to the proposed solution in more detail.
- While the theoretical result in Appendix A seems relatively straightforward and following from existing results (see the question below), its importance is repeatedly mentioned many times in the text (e.g., lines 78, 125, 206, 247).
- Code for reproducing the results is not provided (if I am not mistaken).

Minor: the first parts of the paper assume that only nodes of a certain type have features. The fact that it is possible to apply the method in a general setup is discussed in L360-362. I advise moving this explanation to earlier in the text.

Typos:
- L143: \citet should be \cite
- L220: one of V_k^d should be V_u^d
- L376: "we apply missing to a feature matrix"
- L1102: "highest.Since"

**Questions:**

Q1. I could not find in the text the details on how the models are evaluated for link prediction (dataset splits, negative sampling, details for metric computation, etc.) Could you please provide the details or point to a particular part of the text?

Q2. Regarding the theoretical analysis in Appendix A, it is written in line 725 "Here, we prove the case when non-attributed nodes participate in feature diffusion with virtual features." Do virtual features bring additional challenges to the proof? It seems that you treat nodes with missing features and with no features exactly the same (in line 735), so it reduces to the standard scenario.

---

> ### Author Response · Authors · 2024-11-24
> **Response to Reviewer QsiT (1/3)**
>
> We sincerely appreciate the reviewer’s thorough evaluation and detailed feedback, which offer valuable insights and help us identify ways to enhance our work. We provide our responses below.
>
> >**W1.** It seems that there are two assumptions needed for the approach to work: 1) the graph should be homophilic (nodes with small graph distances between them have similar features) so that feature propagation works, and 2) features are of the same nature/scale so that equation (5) is meaningful. These limitations should be discussed.
>
> **A1.**
>
> * 1$)$ the graph should be homophilic
>
> **In heterogeneous graphs, some node types often lack features, making it very challenging to assess the overall level of feature homophily in such graphs** (i.e., whether the graph is feature-homophilic or non-feature-homophilic (feature-heterophilic)). Nevertheless, if one wishes to measure feature homophily, it can be done using metapaths (e.g., Movie-Director-Movie or Movie-Actor-Movie) that connect node types with features. Each **metapath exhibits a distinct level of feature homophily within a heterogeneous graph**. For example, some metapaths, such as Movie-Director-Movie, exhibit high feature homophily, while others, like Movie-Actor-Movie, display low feature homophily.
>
> Table 17 in the revised manuscript presents the normalized Dirichlet energy (lower values indicate higher feature homophily), representing the feature homophily levels for each metapath in the datasets used in this paper. As shown in the table, all datasets contain metapaths with varying levels of feature homophily. We emphasize that, **while it is very challenging to determine whether a heterogeneous graph is feature-homophilic or feature-heterophilic, diverse levels of feature homophily exist within the graph depending on the metapaths**. Through the extensive experiments, we confirm that our HetGFD effectively prevents performance degradation on downstream tasks under various missing data scenarios across these datasets as well as the PPI dataset.
>
> However, **as the reviewer pointed out, on feature-heterophilic heterogeneous graphs, where most paths connecting attributed nodes are heterophilic connections, the effectiveness of HetGFD may be limited**.
>
> **Table 17**: https://anonymous.4open.science/r/ICLR_2025_6591_tables-E865/Table%2017.png
>
> * 2$)$ features are of the same nature/scale
>
> Since missing features are imputed through iterative aggregation using a weighted sum of neighboring features, **it is crucial that the features involved in the diffusion process have the same nature and scale**. If the features differ in nature or scale, diffusion-based imputation methods, including HetGFD, are unlikely to achieve effective imputation. Therefore, the assumption regarding the uniformity of feature nature and scale is essential for the efficacy of diffusion-based imputation methods.
>
>
> **We have incorporated a discussion of these limitations into the conclusion** section of the revised manuscript. Furthermore, **we have included this important discussion regarding feature homophily in Appendix E** of the revised manuscript
>
> >**W2.** Some of the components of the proposed solution are designed based on some intuition, but particular formulas are not well motivated (e.g., by models or theoretical analysis). Examples include a particular weighting of the edge types in eq. (2), ranking edge types according to homophily and eq. (6), and equation (8). I acknowledge the ablation studies in Appendix, but still many potential solutions could be used and it would be insightful to understand why a particular form is chosen.
>
> We understand the reviewer’s concern regarding the need for more detailed explanations behind the design choices and formulas in the manuscript. In response, **we have thoroughly clarified the motivations for key components of our method**, including the weighting of edge types in Eq. (2), the ranking of edge types based on homophily in Eq. (6), and the formulation in Eq. (8). **These revisions are included in the proposed method section of the revised manuscript** and aim to provide a clearer rationale for our design decisions. We sincerely thank the reviewer for taking the time to deeply understand our work despite the limited explanations in the original manuscript. **We are confident that if the reviewer revisits the proposed method section, they will find it significantly much easier to understand**.

---

> ### Author Response · Authors · 2024-11-24
> **Response to Reviewer QsiT (2/3)**
>
> >**W3.** One of the baselines (PCFI+VF) shows comparable performance in most cases, I think it would be useful to discuss this method and compare its advantages and disadvantages to the proposed solution in more detail.
>
> **A3.** The main difference between PCFI+VF and HetGFD is that **PCFI+VF cannot take into account any information inherent in edge types on heterogeneous graphs**. Instead, PCFI+VF treats all edges equally, regardless of their types. In contrast, HetGFD controls the overall diffusion process at both the feature level (i.e., PC) and the edge level (i.e., $\beta^{k-1}$ in Eq. (8)) by incorporating edge type information. To address the reviewer’s concern, **we have restructured the proposed method section to emphasize the comparison with PCFI**, as shown in lines 302-306 and lines 330-342 of the revised manuscript. Furthermore, **we have discussed the significant gap between PCFI and HetGFD** in lines 364-371 of the revised manuscript.
>
>
> >**W4.** While the theoretical result in Appendix A seems relatively straightforward and following from existing results (see the question below), its importance is repeatedly mentioned many times in the text (e.g., lines 78, 125, 206, 247).
>
> **A4.** We did not intend to emphasize the theoretical result itself but rather to address potential concerns about the validity of zero initialization. In the revised manuscript, **we have removed these mentions from all parts except for line 247**.
>
> >**W5.** Code for reproducing the results is not provided (if I am not mistaken).
>
> **A5.** We have made the **code for HetGFD publicly available** at the following link.
>
> **Code**: https://anonymous.4open.science/r/ICLR_6591_code-AA8A/
>
> >**Q1.** I could not find in the text the details on how the models are evaluated for link prediction (dataset splits, negative sampling, details for metric computation, etc.) Could you please provide the details or point to a particular part of the text?
>
> **A6.** **We have revised the manuscript to include separate subsections for the experimental details of semi-supervised node classification (Appendix B.3) and link prediction (Appendix B.4)**, replacing the previous combined explanation. Furthermore, **we have added the missing details regarding negative sampling and metric computation for link prediction** in the revised manuscript.
>
> * Negative sampling
>
> First, we create a mask that identifies all potential edges in the graph, **excluding existing edges to form a pool** of non-edges. From this pool, **we randomly sample negative edges**, ensuring the number of negative samples matches the desired ratio for validation and testing. The remaining non-edges are used to create a training mask for negative samples.
>
> * Metric Computation
>
> **We utilize the ROC AUC metric (denoted as AUC) and AP, two metrics widely used for link prediction tasks**.
>
> The ROC AUC metric (denoted as AUC) evaluates the model's ability to distinguish between positive (true) links and negative (non-existent) links. It is computed by assessing the True Positive Rate (TPR) and False Positive Rate (FPR) at various thresholds of the predicted edge scores. These values are used to construct a Receiver Operating Characteristic (ROC) curve, which plots FPR on the x-axis and TPR on the y-axis. The Area Under the Curve (AUC) is then calculated, providing a single value that represents the overall performance of the model.
>
> The AP (Average Precision) metric emphasizes the precision-recall tradeoff, making it particularly valuable for imbalanced datasets. It is calculated by measuring Precision and Recall values across different thresholds of the predicted edge scores. These values are used to create a Precision-Recall curve, and the AP score is derived as the weighted average of precision values at each level of recall. A higher AP score reflects better performance, especially in identifying true links among a large number of negatives.
>
> >**Q2.** Regarding the theoretical analysis in Appendix A, it is written in line 725 "Here, we prove the case when non-attributed nodes participate in feature diffusion with virtual features." Do virtual features bring additional challenges to the proof? It seems that you treat nodes with missing features and with no features exactly the same (in line 735), so it reduces to the standard scenario.
>
> **A7.** For non-attributed nodes, features are absent rather than missing, which is why we specifically generate virtual features for them. **Since virtual features are not known or observed features, they are treated as missing features, allowing the scenario to reduce to the standard case**, as the reviewer mentioned.

---

> ### Author Response · Authors · 2024-11-24
> **Response to Reviewer QsiT (3/3)**
>
> > Minor: the first parts of the paper assume that only nodes of a certain type have features. The fact that it is possible to apply the method in a general setup is discussed in L360-362. I advise moving this explanation to earlier in the text.
>
> **A8.** Based on the reviewer’s advice, we have relocated the explanation regarding the general applicability of the method from lines 360–362 to lines 202–206 in the revised manuscript.
>
> > Typos
>
> **A9.** We have corrected all the typos you mentioned in the revised manuscript. We appreciate your attention to detail, which has helped improve the overall quality of our paper.

---

> ### Comment · Area_Chair_HUP6 · 2024-11-25
>
> Could please acknowledge and respond to the rebuttal.

---

> ### Author Response · Authors · 2024-11-27
> **Eager for Your Feedback on Our Rebuttal**
>
> Dear Reviewer QsiT,
>
> We sincerely thank the reviewer for taking the time to review our work and for the thoughtful evaluation. Your detailed feedback has significantly improved the quality of our work. With only 17 hours remaining for the PDF revision, we are eager to engage further and understand whether our responses have satisfactorily addressed your concerns.
>
> In our rebuttal, **we provided point-by-point responses to all your questions and concerns regarding (1) limitations, (2) motivations for formulas, (3) comparison with PCFI, (4) redundant mentions, (5) code, (6) experimental details on link prediction, and (7) theoretical analysis**. In summary,
>
> * For (1), we have incorporated a discussion of **these limitations into the conclusion section** of the revised manuscript.
> * For (2), **we have thoroughly restructured the proposed method section**.
> * For (3), we clarified that **PCFI+VF cannot account for any information inherent in edge types** on heterogeneous graphs.
> * For (4), **we have removed these mentions** from all parts of the manuscript except for line 247 in the revised version.
> * For (5),  **we have made the code for HetGFD publicly available** at the following link: https://anonymous.4open.science/r/ICLR_6591_code-AA8A/
> * For (6), **we have added the missing details** regarding negative sampling and metric computation for link prediction in the revised manuscript.
> * For (7), **we agree with Reviewer QsiT’s intuition**.
>
> We would greatly appreciate it if you could kindly review our responses. We welcome any further questions and are happy to provide additional clarifications if needed. Thank you for your consideration.
>
> Sincerely,\
> The Authors

---

> > ### Comment · Reviewer_QsiT · 2024-11-29
> >
> > Thank you very much for your detailed responses! My questions and concerns have been addressed. I reviewed the updated paper: the presentation has been improved and, importantly, the potential limitations of the approach are clearly mentioned. I appreciate the authors' efforts in addressing the comments of the reviewers and I maintain a positive assessment of this work.

---

> > > ### Author Response · Authors · 2024-11-30
> > >
> > > Thank you very much for your thoughtful feedback and positive assessment of our work. We sincerely appreciate your recognition of the improvements in the revised paper. Your detailed feedback has significantly contributed to the enhancement of our paper.
> > >
> > > If there are any remaining concerns, please do not hesitate to inform us; we are ready to address them promptly.

---

### Official Review · Reviewer_56Ce · 2024-11-03

**Soundness:** 3
**Presentation:** 2
**Contribution:** 2
**Rating:** 6
**Confidence:** 3

**Summary:**

The paper proposes a novel imputation method named HetGFD. It is designed for heterogeneous graphs with partially observed features. By utilizing a virtual feature mechanism and introducing edge-type-wise homophily, HetGFD enables feature diffusion across different node types and edge connections. The method shows improvements in semi-supervised node classification and link prediction tasks, even under high missing feature rates.

**Strengths:**

S1. HetGFD introduces virtual features for nodes with missing attributes. This approach effectively bridges gaps in the feature set, enhancing the model's robustness.

S2. The integration of edge-type-wise homophily allows for a flexible adjustment of edge weights during feature propagation. It improves the quality of the imputed features.

S3. The paper demonstrates the effectiveness of HetGFD through experiments on multiple benchmark datasets. The results show that HetGFD outperforms traditional imputation methods, highlighting its practical applicability in real-world scenarios.

**Weaknesses:**

W1. The novelty of this paper is limited. The main innovation lies in assigning different propagation weights based on node similarity, which is similar to the k-NN imputation method.

W2. Calculating edge-type weights requires complete pairwise feature distance calculations among all attributed nodes, leading to high computational costs that could limit scalability.

W3. Using incomplete features for similarity calculations makes the results less reliable.

**Questions:**

Q1. Can the authors provide additional experimental results to show that the proposed method outperforms k-NN? The ablation study suggests similar performance, which raises concerns about its novelty.

Q2. The fine-grained similarity calculation method raises questions. Additional discussions and experimental validations regarding the algorithm's efficiency and scalability would be beneficial. Could the authors also demonstrate the effectiveness of a coarser propagation weight calculation method? For instance, using the difference between the feature means of the specific set of attribute nodes and the overall feature means of all attribute nodes could provide a simpler and more efficient approach. Additionally, Section D3 does not fully outline the model's complexity, particularly regarding similarity and shortest path computations.

Q3. Could the authors provide further motivation or theoretical guarantees, such as lower-bound proofs, to validate the similarity calculation strategy for incomplete features? Additionally, using attention mechanisms or learnable weight matrices for dynamic adjustment of propagation weights for the specific downstream task may be a more effective approach. The authors can further discuss the motivation behind their strategy.

---

> ### Author Response · Authors · 2024-11-24
> **Response to Reviewer 56Ce (1/2)**
>
> We sincerely thank the reviewer for their thoughtful evaluation and constructive feedback, which have significantly improved our work. We provide our responses below.
>
> >**W1.** The novelty of this paper is limited. The main innovation lies in assigning different propagation weights based on node similarity, which is similar to the k-NN imputation method.\
> **Q1.** Can the authors provide additional experimental results to show that the proposed method outperforms k-NN? The ablation study suggests similar performance, which raises concerns about its novelty.
>
> **A1.** HetGFD and k-Nearest-Neighbors (kNN) imputation [1] may appear similar, as both leverage feature similarity for imputation. However, **there is a fundamental difference between the two**. HetGFD utilizes the structure of the given heterogeneous graph, whereas kNN imputation does not consider the graph structure. kNN performs imputation by creating $k$ edges for each node based on cosine similarity and leveraging the neighborhood mean. In other words, **kNN forms a graph structure to perform imputation but does not use an existing graph structure**. Since kNN imputation is a widely used method across various domains, **we have included it as a baseline for all experiments to ensure a comprehensive comparison**, as demonstrated in Figure 4, Figure 9, Table 1, Table 2, Table 3, and Table 18 of the revised manuscript. As shown in the figures and tables, kNN imputation demonstrates similar performance to zero imputation and mean imputation, while **HetGFD consistently outperforms kNN imputation by a significant margin**.
>
> [1] Troyanskaya, Olga, et al. "Missing value estimation methods for DNA microarrays." Bioinformatics 17.6 (2001): 520-525.
>
> >**W2.** Calculating edge-type weights requires complete pairwise feature distance calculations among all attributed nodes, leading to high computational costs that could limit scalability.
>
> **A2.** The calculation of feature similarity in Eq. (5) might seem computationally expensive. However, **it has a time complexity of $O(F \cdot N)$**, where $F$ denotes the number of features and $N$ denotes the number of nodes. **This computation takes only 0.0327 seconds on the DBLP dataset with $N=26128$ and $F=4231$** when performed on a single GPU (NVIDIA GeForce RTX 2080 Ti). If the network size increases further, the computational complexity can be reduced by sampling a fixed number of edge pairs for computation, similar to how the average similarity of random pairs is calculated in the denominator of Eq. (5).
>
> >**W3.** Using incomplete features for similarity calculations makes the results less reliable.
>
> **A3.** **We do not perform the calculation of feature similarity on incomplete features** containing missing values. **To avoid this** calculation on incomplete features, **we position preliminary diffusion as the first step of HetGFD**, as shown in Figure 2 of the manuscript. Preliminary diffusion generates a pre-imputed feature matrix using Eq. (4). **From this imputed complete matrix, feature similarity is calculated** to derive edge-type-wise homophily, as described in Eq. (5).

---

> ### Author Response · Authors · 2024-11-24
> **Response to Reviewer 56Ce (2/2)**
>
> >**Q2-1.** The fine-grained similarity calculation method raises questions. Additional discussions and experimental validations regarding the algorithm's efficiency and scalability would be beneficial. Could the authors also demonstrate the effectiveness of a coarser propagation weight calculation method? For instance, using the difference between the feature means of the specific set of attribute nodes and the overall feature means of all attribute nodes could provide a simpler and more efficient approach.
>
> **A4.** **We conduct additional experiments to compare the weighted matrix $\mathbf{\bar{W}}^{(d)}$** used in relation-aware diffusion **against feature distance-based weights**. The purpose of $\mathbf{\bar{W}}^{(d)}$ is to assign greater weights to edge types that significantly contribute to feature homophily, *i.e.*, edge types that result in highly similar features between connected nodes. In contrast, the feature distance-based approach can be a simpler method for calculating weights. To construct the feature distance-based propagation matrix, we follow these steps: First, a pre-imputed feature matrix is generated using preliminary diffusion, as in HetGFD. Then, a single feature mean vector is computed for each node type based on the pre-imputed matrix. For each edge type, the feature distance is calculated by measuring the distance between the mean feature vectors of the two node types connected by that edge type. For each edge type, the calculated distance is inverted and assigned to the edges belonging to that edge type. Finally, the resulting weighted matrix is row-stochastically normalized to perform diffusion-based imputation.
>
> Table 15 in the revised manuscript presents a performance comparison between the feature distance-based approach and our HetGFD. For this comparison, we perform semi-supervised classification and link prediction tasks, consistently utilizing HGT models as downstream GNNs. As shown in the table, **HetGFD significantly outperforms the feature distance-based approach across all settings**. These results **validate the effectiveness of designing the propagation matrix based on edge-type-wise homophily $\mathcal{H}$**.
>
> **Table 15**: https://anonymous.4open.science/r/ICLR_2025_6591_tables-E865/Table%2015.png
>
> **Q2-2.** Additionally, Section D3 does not fully outline the model's complexity, particularly regarding similarity and shortest path computations.
>
> **A5.** To address the reviewer’s concern regarding HetGFD’s complexity, **we have conducted an in-depth analysis by examining each process** that comprises HetGFD.
>
> The main processes to consider in the time complexity of HetGFD are the two diffusion stages, the calculation of edge-type-wise homophily $\mathcal{H}$ the calculation of PC. The two diffusion stage have a time complexity of $O(|\mathcal{E}|)$ since they are implemented using the message passing operation in PyTorch Geometric [2]. This operation involves aggregating information from neighboring nodes, which scales linearly with the number of edges in the graph. The calculation of $\mathcal{H}$ has a time complexity of $O(F\cdot|\mathcal{E}|)$ since it measures feature similarity across all edges. The calculation of PC has a time complexity of $O(N^2)$. This is because the calculation of PC uses Dijkstra's algorithm, an algorithm for finding the shortest paths between nodes in a graph, for distance encoding. In structural-missing settings, the missing status of nodes is identical across all channels. Therefore, only a single transition matrix needs to be computed, requiring just one PC calculation. However, in uniform-missing settings, the missing status of nodes varies for each channel, necessitating different transition matrices for each of the $F$ channels. Consequently, $F$ PC calculations are required. Thus, HetGFD operates **in structural-missing settings with a time complexity of $O(F\cdot|\mathcal{E}|+N^2)$** and **in uniform-missing settings with a complexity of $O(F\cdot|\mathcal{E}|+F \cdot N^2)$**. However, it is important to note that **PCFI, the most competitive baseline, also involves the calculation of PC using Dijkstra's algorithm, which has a time complexity of $O(N^2)$**. Compared to PCFI, **our HetGFD consistently demonstrates superiority** across all experimental settings.
>
> We have included a detailed explanation of HetGFD's complexity in Appendix D.3 of the revised manuscript.
>
> [2] Fey, Matthias, and Jan Eric Lenssen. "Fast graph representation learning with PyTorch Geometric." arXiv preprint arXiv:1903.02428 (2019).

---

> ### Comment · Area_Chair_HUP6 · 2024-11-25
>
> Could please acknowledge and respond to the rebuttal.

---

> > ### Comment · Reviewer_56Ce · 2024-11-27
> >
> > Thank you for your response. However, I still have concerns about the novelty and scalability of the method, so I will keep my score.

---

> ### Author Response · Authors · 2024-11-27
> **Additional Clarifications on Novelty and Scalability**
>
> Thank you for your thoughtful feedback and for taking the time to review our work. We understand your remaining concerns regarding the novelty and scalability of our method, and provide additional clarifications and evidence to address these points as below.
>
> * Novelty: Our HetGFD is fundamentally different from kNN imputation [1], which Reviewer 56Ce identified as a similar method. **We clarified that, unlike kNN imputation, which does not utilize the graph structure of the given heterogeneous graph, HetGFD explicitly leverages the graph structure**. Furthermore, we emphasize that our novelty lies in: **(1) the introduction of virtual features**, which enable propagation through non-attributed nodes, and **(2) relation-aware diffusion**, which facilitates the use of information inherent in edge types. To the best of our knowledge, this work is **the first attempt to leverage diffusion-based feature imputation for heterogeneous graphs**.
>
> * Scalability: We discussed the time complexity of HetGFD and clarified that **PCFI, the most competitive baseline, also has a time complexity of $O(N^2)$**. To address Reviewer 56Ce’s remaining concern regarding scalability, we further conduct experiments on the OGBN-Arxiv dataset [2], which consists of 169,343 nodes and 1,166,243 edges. **Despite the size of this dataset, our method required only 23.5 minutes for imputation** with the structural-missing setting with a missing rate of 99.5\%, showcasing its efficiency. We have included this experimental result and further details, demonstrating the scalability of HetGFD, in Appendix D.3.
>
> We hope this additional explanation addresses your remaining concerns. **If you have any further concerns, we would be happy to address them**. Thank you again for your valuable comments.
>
> [1] Troyanskaya, Olga, et al. "Missing value estimation methods for DNA microarrays." Bioinformatics 17.6 (2001): 520-525.\
> [2] Hu, Weihua, et al. "Open graph benchmark: Datasets for machine learning on graphs." Advances in neural information processing systems 33 (2020): 22118-22133.

---

> ### Author Response · Authors · 2024-12-02
> **[Gentle Reminder] Kindly Seeking Your Feedback on Our Clarifications Before the Deadline**
>
> Dear **Reviewer 56Ce**,
>
> In response to Reviewer 56Ce's remaining concerns, we provided detailed clarifications to address them. With only **21 hours remaining before the deadline**, we are eager to confirm whether our clarifications have sufficiently addressed these concerns. We kindly ask you to take a moment to review our clarifications and share any additional feedback. If there are any further questions or concerns, please rest assured that we are fully prepared to respond promptly.
>
> Thank you once again for your professional feedback, which has greatly enhanced our paper.
>
> Sincerely,\
> The Authors

---

### Official Review · Reviewer_wM4n · 2024-11-05

**Soundness:** 3
**Presentation:** 3
**Contribution:** 2
**Rating:** 5
**Confidence:** 3

**Summary:**

Authors provide the framework that can diffuse missing feature values in the heterogeneous graphs. Authors divide nodes into 3 groups for each feature channel -- nodes with known features, nodes with unknown features, and non-attributed nodes (virtual features) -- and then define the diffusion across those groups along with graph links. In this diffusion process, authors introduce 3 kinds of weights: (1) normalization by the frequency of links, (2) edge-type aware weights based on edge-type homophily, and (3) pseudo-confidence aware weights. After multiple steps of diffusion, the diffused features for attributable nodes are obtained by removing the virtual features of the non-attributed nodes.

Authors conduct experiments for semi-supervised node classification as well as link prediction tasks by applying the diffused features into a common GNN algorithm. From the empirical experimental results, authors provide the superior performance from the proposed method as opposed to the baselines.

**Strengths:**

- The idea of virtual features to allow the diffusion over heterogeneous graphs is novel and worth studying.
- Authors show the superior performance of the proposed algorithm for structural as well as random missing features. And its gain increases as the missing rate increases.
- The proposed algorithm is straightforward.

**Weaknesses:**

- There is almost no learning process and the proposed method is closer to the well-designed preprocessing process. Some model learning from data would fit better to the venue.
- There are crucial hyperparameters, but setting them up is decoupled from the task learning. Sensitivity analysis and the proposal for tuning them would be great to have.
- Homophily is assumed for the proposed diffusion process, while the underlying GNN could capture more general interactions. It would be great to have experiments on the non-homophily dataset.

**Questions:**

- In Eq (5), how can the homophily be defined when the edge type r connects two different types of nodes? For instance, if r connects actor and movie type, then actor node is the non-attributable node for the move type feature. How can we define the similarity here?

- For the link prediction task, applying the diffusion process should be careful since using the links in the test split for the diffusion path is information leakage. Despite this demand for carefulness, it is not addressed in the paper. If authors have not done this way, experiments should be performed in this way. If authors already performed this way, the procedure should be clearly addressed.

- It would be great to see the ablation study such as the impact of convergence (i.e. k steps in diffusion), the importance of edge type ranking , and so on.

---

> ### Author Response · Authors · 2024-11-24
> **Response to Reviewer wM4n (1/3)**
>
> We sincerely thank the reviewer for their constructive feedback and valuable suggestions, which have greatly improved our work. Our responses are provided below.
>
> >**W1.** There is almost no learning process and the proposed method is closer to the well-designed preprocessing process. Some model learning from data would fit better to the venue.
>
> **A1.** As the reviewer mentioned, HetGFD can be viewed as a preprocessing process. However, as stated in lines 177-178, **the problem we address is ensuring effective learning from data with missing features**. In other words, while what we design is an imputation method, the overall framework we propose includes GNNs learning from data. **The reason why we choose a non-learning-based approach is to design an imputation method that can be applied to various GNNs**. This is particularly important because there are many types of GNNs and the field is continuously evolving. Furthermore, HetGFD is not specifically designed for a particular task, yet it demonstrates outstanding performance in both node classification and link prediction.
>
> In summary, **our work falls under representation learning in settings with missing features, targeting graph-related tasks**. As a recent study proposing a diffusion-based imputation method for homogeneous graphs was presented at ICLR 2023, we strongly believe that our research is well-suited for this venue.
>
> >**W2.** There are crucial hyperparameters, but setting them up is decoupled from the task learning. Sensitivity analysis and the proposal for tuning them would be great to have.
>
> **A2.** **In fact, we have already conducted an analysis of hyperparameter sensitivity in Appendix C** of the manuscript. As demonstrated in Figures 5, 6, and 7, HetGFD models with most combinations of $\alpha$ and $\beta$ outperform the existing state-of-the-art performance in each setting. Table 8 in the manuscript presents the performance of HetGFD with varying values of $K$, the number of diffusion steps. While we consistently use $K=100$ across all experiments involving HetGFD, the results indicate that HetGFD with $K \geq 40$ demonstrates robustness in downstream tasks. As  shown in Table 5, Table 6, and Table 7 of the manuscript, the optimal $(\alpha, \beta)$ values differ depending on the specific setting. Thus, when tuning HetGFD, **we recommend setting $K=100$ and searching for $(\alpha, \beta)$ within the range of {($\alpha, \beta)|\alpha \in$ {$0.9, 0.7, 0.5, 0.3, 0.1$}, $\beta  \in$ {$0.99, 0.9, 0.8, 0.5, 0.4, 0.2, 0.1, 0.05$}}** based on validation sets, as used in this paper.
>
> We realize that the hyperparameter search range could be inferred from Figures 5, 6, and 7 but was not explicitly stated in the original manuscript. We have added the hyperparameter search range in lines 943-945 of the revised manuscript.

---

> ### Author Response · Authors · 2024-11-24
> **Response to Reviewer wM4n (2/3)**
>
> >**W3.** Homophily is assumed for the proposed diffusion process, while the underlying GNN could capture more general interactions. It would be great to have experiments on the non-homophily dataset.
>
> **A3.** In the context of homophily, two types are commonly discussed: class homophily and feature homophily. As the reviewer noted, diffusion-based imputation methods rely on feature homophily. **However, in heterogeneous graphs, some node types often lack features, making it very challenging to assess the overall level of feature homophily in such graphs** (i.e., whether the graph is feature-homophilic or non-feature-homophilic (feature-heterophilic)). Nevertheless, if one wishes to measure feature homophily, it can be done using metapaths (e.g., Movie-Director-Movie or Movie-Actor-Movie) that connect node types with features. Each **metapath exhibits a distinct level of feature homophily within a heterogeneous graph**. For example, some metapaths, such as Movie-Director-Movie, exhibit high feature homophily, while others, like Movie-Actor-Movie, display low feature homophily.
>
> Table 17 in the revised manuscript presents the normalized Dirichlet energy (lower values indicate higher feature homophily), representing the feature homophily levels for each metapath in the datasets used in this paper. As shown in the table, all datasets contain metapaths with varying levels of feature homophily. We emphasize that, **while it is very challenging to determine whether a heterogeneous graph is feature-homophilic or feature-heterophilic, diverse levels of feature homophily exist within the graph depending on the metapaths**. Through the extensive experiments, we confirm that our HetGFD effectively prevents performance degradation on downstream tasks under various missing data scenarios across these datasets as well as the PPI dataset.
>
> **However, on feature-heterophilic heterogeneous graphs, where most paths connecting attributed nodes are heterophilic connections, the effectiveness of HetGFD may be limited**. To the best of our knowledge, there is no existing benchmark dataset for feature-heterophilic heterogeneous graphs. If the reviewer is aware of a feature-heterophilic heterogeneous graph dataset, we would greatly appreciate the suggestion and would promptly conduct experiments using it. **We have added this limitation of HetGFD to the conclusion section** and have included this important discussion regarding feature homophily in Appendix E of the revised manuscript.
>
> **Table 17**: https://anonymous.4open.science/r/ICLR_2025_6591_tables-E865/Table%2017.png

---

> ### Author Response · Authors · 2024-11-24
> **Response to Reviewer wM4n (3/3)**
>
> >**Q1.** In Eq (5), how can the homophily be defined when the edge type r connects two different types of nodes? For instance, if r connects actor and movie type, then actor node is the non-attributable node for the move type feature. How can we define the similarity here?
>
> **A4.** As in the example provided by the reviewer, let us consider the IMDB dataset, where actor nodes are non-attributed nodes, and only movie nodes possess features as attributed nodes. As the reviewer correctly pointed out, if $r$ connects actor and movie type, feature similarity cannot be directly computed due to the absence of features in actor nodes. **This is the reason behind the design of preliminary diffusion**. Before the preliminary diffusion, for non-attributed nodes, we assign virtual features filled with zeros for every non-attributed node, where the dimension of the virtual features is set to match the feature dimension of the attributed nodes. These virtual features for non-attributed nodes, along with the missing features of attributed nodes, are iteratively updated during the preliminary diffusion. This stage outputs a pre-imputed feature matrix in which both virtual features and missing features are updated. **Consequently, on the pre-imputed feature matrix, feature similarity can then be computed** using Eq. (5) in the manuscript.
>
> >**Q2.** For the link prediction task, applying the diffusion process should be careful since using the links in the test split for the diffusion path is information leakage. Despite this demand for carefulness, it is not addressed in the paper. If authors have not done this way, experiments should be performed in this way. If authors already performed this way, the procedure should be clearly addressed.
>
> **A5.** We appreciate the reviewer for raising this important concern about potential information leakage during the diffusion process for the link prediction task. We confirm that in all our experiments, the diffusion process is carefully applied to avoid any information leakage. Specifically, **we perform both imputation and the training of downstream GNN models only on the training edges in each split, strictly excluding edges in the validation and test sets**. This ensures that no information from the test set is used during the diffusion process. We have observed that using testing edges during the imputation process results in AP and ROC AUC scores approaching 100%, highlighting the importance of carefully excluding test edges to ensure fair evaluation.
> We realize that this critical detail was not explicitly stated in the original manuscript. To prevent any misunderstanding, we have added a clear explanation of this procedure in lines 888–890 of the revised manuscript.
>
> >**Q3.** It would be great to see the ablation study such as the impact of convergence (i.e. k steps in diffusion), the importance of edge type ranking , and so on.
>
> **A6.** **In fact, we have already conducted a comprehensive ablation study in Appendix D.1** of the manuscript. This study evaluates **the impact of various factors, including convergence (i.e., the number of diffusion steps, $K$), PC, edge-type ranking, zero initialization, preliminary diffusion, and the transition matrix**. We hope this section addresses the reviewer's concern, and we are happy to elaborate further if needed.

---

> > ### Author Response · Authors · 2024-12-02
> > **[Gentle Reminder] Kindly Seeking Your Feedback on Our Rebuttal Before the Deadline**
> >
> > Dear **Reviewer wM4n**,
> >
> > In our rebuttal, we provided detailed, point-by-point responses to all the concerns and questions you raised. **However, we have yet to receive any response from you**. With only **21 hours remaining before the deadline**, we are eager to confirm whether our responses have sufficiently addressed your concerns. We strongly believe that our work has been significantly improved by incorporating your feedback, and we are pleased to note that **all other reviewers have expressed opinions supportive of its acceptance.**
> >
> > We kindly ask you to take a moment to review our rebuttal and share any feedback. If you have any remaining questions or concerns, please do not hesitate to let us know, and we will address them promptly.
> >
> > Sincerely,\
> > The Authors

---

> ### Comment · Area_Chair_HUP6 · 2024-11-25
>
> Could please acknowledge and respond to the rebuttal.

---

> ### Author Response · Authors · 2024-11-27
> **Eager for Your Feedback on Our Rebuttal**
>
> Dear Reviewer wM4n,
>
> We sincerely thank you for dedicating your time to review our work and for providing thorough feedback. Your insights have significantly contributed to improving the quality of our paper. With only 17 hours remaining for the PDF revision, we are eager to engage further and understand if our responses have satisfactorily addressed your concerns.
>
> In our rebuttal, **we provided point-by-point responses to all your questions and concerns regarding (1) hyperparameter sensitivity analysis, (2) the ablation study, (3) the non-learning-based approach, (4) feature homophily, (5) defining similarity using non-attributed nodes, and (6) the link prediction setting**. In summary:
>
> * For (1), the original manuscript **already included** an analysis of hyperparameter sensitivity in Appendix C.
> * For (2), the original manuscript **already included** a comprehensive ablation study in Appendix D.1.
> * For (3), we clarified that the reason we chose a non-learning-based approach was to design an imputation method that **can be applied to various GNNs for both node classification and link prediction**.
> * For (4), we conducted **an in-depth discussion on feature homophily** in heterogeneous graphs.
> * For (5), we clarified that **we do not define similarity using incomplete features**.
> * For (6), we clarified that we **strictly followed standard experimental settings** for link prediction.
>
> We would greatly appreciate it if you could kindly review our responses. We welcome any further questions and are happy to provide additional clarifications if needed. Thank you for your consideration.
>
> Sincerely,\
> The Authors

---

> ### Author Response · Authors · 2024-12-03
> **[Gentle Reminder] Eager for Your Feedback on our Rebuttal**
>
> Dear **Reviewer wM4n**,
>
> With **only one hour remaining before the deadline**, we are eager to confirm whether our responses have adequately addressed your concerns. We kindly request you to take a moment to review our rebuttal and share any feedback.
>
> Sincerely,\
> The Authors

---

> > ### Comment · Area_Chair_HUP6 · 2024-12-03
> >
> > Reviewer wM4n, could you please update your initial review with a reply to the authors rebuttal? Thanks.

---

### Author Response · Authors · 2024-11-24
**General Response**

This paper proposes Heterogeneous Graph Feature Diffusion (HetGFD), a novel imputation scheme that enables diffusion-based imputation in heterogeneous graphs. HetGFD tackles the significant challenges of applying diffusion-based imputation to heterogeneous graphs, achieving outstanding performance in semi-supervised node classification and link prediction across various feature-missing settings.

**We express our profound gratitude to all reviewers for dedicating their time and effort to thoroughly evaluate our manuscript. The constructive feedback provided by the reviewers has greatly enhanced the quality of our paper**. To address the reviewers’ concerns, **we provide point-by-point responses to all the questions and concerns raised by the reviewers**. Below is a summary of the key updates made to our paper:
* Reorganization of the Proposed Method Section (in Sec. 4)
* The Inclusion of kNN Imputation as a Baseline (in Sec. 5)
* Statistical Analysis (in Appendix D.5)
* Discussion on Feature Homophily (in Appendix. E)
* Complexity Analysis (in Sec. 6)
* Additional Complexity Analysis (in Appendix D.3)
* Limitations (in Sec. 7)
* Experimental Details for Link Prediction (in Appendix B.4)
* Hyperparameter Search Range (in lines 943-945 of Appendix. B.6)
* Comparison with Feature Distance-Based Weights (in Appendix. C.11)
* Code: https://anonymous.4open.science/r/ICLR_6591_code-AA8A/

**For the revised and updated parts in the manuscript, we have marked the changes in blue**. We hope that the responses provided below address all the reviewers’ concerns, and **please let us know if the raised concerns are addressed**. We are happy to answer further questions and would be delighted to provide additional clarifications.

---

### Author Response · Authors · 2024-12-03
**Final Summary**

Dear Senior Area Chairs, Area Chairs, and Reviewers,

We would like to express our gratitude to both the chairs and the reviewers for their hard work. During the discussion period, the reviewers provided thorough and professional feedback, enabling us to significantly enhance the quality of the paper by addressing all their concerns and incorporating their suggestions. **Except for Reviewer wM4n, all reviewers have expressed opinions supportive of acceptance**.

**Despite our repeated attempts to seek confirmation from Reviewer wM4n, we have not yet received any response over the past two weeks**. To facilitate the discussions among the chairs and reviewers, we would like to summarize (1) the contributions of our work and (2) our rebuttal addressing all of Reviewer wM4n's concerns as a final note. We hope these points will be taken into consideration when the final decision is made.
***
## **Contributions**
1. To the best of our knowledge, this work is **the first attempt to utilize diffusion-based imputation on heterogeneous graphs**.
2. We address significant challenges in heterogeneous graphs to enable feature diffusion: \
2-1) Non-attributed nodes $\rightarrow$ **Virtual feature generation**\
2-2) Various edge types $\rightarrow$ A new measure termed **edge-type homophily, enabling relation-aware distance encoding**
3. Across various missing rates, including **99.5\%**, our HetGFD demonstrates **superiority on various datasets in both semi-supervised node classification and link prediction tasks**.
***
## **Reviewer wM4n's Concerns**
**The absence of a hyperparameter sensitivity analysis**
* We clarified that the original manuscript **already included** an analysis of hyperparameter sensitivity in Appendix C.

**The absence of an ablation study**
* We clarified that the original manuscript **already included** a comprehensive ablation study in Appendix D.1.

**How can we define the similarity using incomplete features?**

  * We clarified that **we do not define similarity using incomplete features**.

**Non-learning-based approach**
  * We clarified that the reason we chose a non-learning-based approach was **to design an imputation method that can be applied to various GNNs for both semi-supervised node classification and link prediction**.

**Link prediction setting**
* We clarified that **we strictly adhered to standard experimental settings for link prediction**.

**Feature homophily**
  * We conducted an **in-depth discussion on feature homophily in heterogeneous graphs** in Appendix E of the revised manuscript.

***
**We have thoroughly addressed Reviewer wM4n's concerns, but we have not received any response**. We strongly believe that the ICLR Senior Area Chairs, Area Chairs, and other reviewers will agree with our perspective upon reviewing our rebuttal to Reviewer wM4n. **We hope that Reviewer wM4n will take the time to review our rebuttal even after the discussion period has concluded**. While we respect Reviewer wM4n's evaluation, we trust that our rebuttal to Reviewer wM4n and the lack of confirmation from Reviewer wM4n will be carefully considered in the final decision-making process.

Once again, we extend our sincerest thanks to all the chairs and reviewers for their efforts.

Sincerely,\
The Authors

---

### Meta-Review · Area_Chair_HUP6 · 2024-12-19

**Metareview:**

This paper proposes a novel method to handle missing data in heterogeneous graphs, where nodes and edges can have different types. By introducing virtual features for undefined nodes and prioritizing edge types during diffusion, the method achieves significant performance improvements in tasks like node classification and link prediction.

All reviewer's concerns have been addressed except of reviewer wM4n who was unresponsive throughout the rebuttal period.
I suggested for acceptance of the paper.

**Additional Comments On Reviewer Discussion:**

see my comments above.

---

### Decision · Program_Chairs · 2025-01-22

Accept (Poster)